# Punctuation-level Attack: Single-shot and Single Punctuation Attack Can Fool Text Models

**Wenqiang Wang**[1]    **Chongyang Du**[1]    **Tao Wang**[2]    **Kaihao Zhang**[3]

**Wenhan Luo**[1*]    **Lin Ma**[4]    **Wei Liu**[5]    **Xiaochun Cao**[1]

[1]Shenzhen Campus of Sun Yat-sen University    [2]Nanjing University
[3]Australian National University    [4]Meituan    [5]Tencent

## Abstract

The adversarial attacks have attracted increasing attention in various fields including natural language processing. The current textual attacking models primarily focus on fooling models by adding character-/word-/sentence-level perturbations, ignoring their influence on human perception. In this paper, for the first time in the community, we propose a novel mode of textual attack, punctuation-level attack. With various types of perturbations, including insertion, displacement, deletion, and replacement, the punctuation-level attack achieves promising fooling rates against SOTA models on typical textual tasks and maintains minimal influence on human perception and understanding of the text by mere perturbation of single-shot single punctuation. Furthermore, we propose a search method named Text Position Punctuation Embedding and Paraphrase (TPPEP) to accelerate the pursuit of optimal position to deploy the attack, without exhaustive search, and we present a mathematical interpretation of TPPEP. Thanks to the integrated Text Position Punctuation Embedding (TPPE), the punctuation attack can be applied at a constant cost of time. Experimental results on public datasets and SOTA models demonstrate the effectiveness of the punctuation attack and the proposed TPPE. We additionally apply the single punctuation attack to summarization, semantic-similarity-scoring, and text-to-image tasks, and achieve encouraging results.

## 1 Introduction

Deep Neural Networks (DNNs) have achieved tremendous success in the NLP community and have spawned a series of well-known applications such as ChatGPT [13]. However, more attention should be paid to the vulnerability of NLP models under adversarial attacks to protect them. Adversarial examples have been shown to have a devastating impact in the image domain [39, 12, 38]. Unlike the image domain, text is a discrete domain [45], which means that any small perturbations, such as sentence, word, or even character perturbations in text, would be easily perceived by humans. It is difficult to achieve imperceptible perturbations. Additionally, the text is a non-differentiable domain, which results in traditional optimization-based methods being ineffective in the NLP domain. Recent research on textual adversarial attacks can be mainly classified into character-level, word-level, sentence-level, and multi-level attacks [41]. Character-level attacks modify words by inserting, deleting, misspelling, replacing, or swapping characters. These attacks are easily detectable by humans and some apps can correct them automatically. However, character-level attacks require multiple queries to determine which characters to attack based on function gradient or score. Word-level attacks modify words by adding, deleting, or replacing important words in a text. Similar to character-level attacks, word-level attacks also require multiple queries to determine which words to

---

*Corresponding Author: < whluo.china@gmail.com >

37th Conference on Neural Information Processing Systems (NeurIPS 2023).

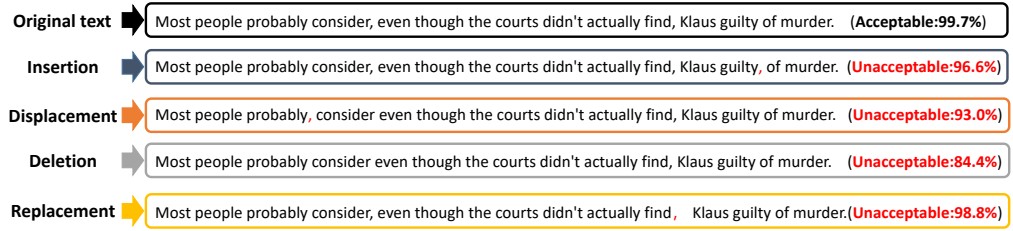

Figure 1: The examples of punctuation-level attacks. The text model is fooled by inserting, displacing, deleting, and replacing punctuation from the original text. The predicted label of the input text is either acceptable or unacceptable. A single punctuation attack can significantly change the score of the predicted label.

attack. Both character- and word-level attacks are iterative and thus may be time-consuming in real NLP applications due to multiple queries required for each attack iteration. Sentence-level attacks modify entire texts by paraphrasing or adding meaningless sentences to input texts. The significant perturbations caused by sentence-level attacks may sometimes change the ground truth of input text. Perturbations in character-, word-, and sentence-levels can cause semantic gaps from the original text and are easily detectable by humans.

Considering the issues mentioned above, we propose punctuation-level attacks as a new method of textual adversarial attack. Compared with other attack methods, perturbations on punctuations are less perceivable than character- and word-level attacks, more difficult to detect than character-level attacks, and cause less impact on ground truth than other types of attacks. In previous research, little attention has been paid to punctuation; several researchers have inserted punctuations into words to create out-of-vocabulary (OOV) words, which is a specific method of character-level attacks. On the contrary, we first propose punctuation-level attacks that regard punctuation perturbations as a new level of attack similar to character-, word-, and sentence-level attacks. We define basic attacking methods and analyze why these methods can fool text functions. Unlike other types of attacks, punctuation-level attacks only modify punctuations by inserting, displacing, deleting, or replacing them without perturbing characters, words, and sentences. Punctuation-level attacks can deceive text functions by inserting, displacing, deleting, or replacing punctuations. Specific examples are presented in Fig. 1, showing that the single punctuation attack results in a dramatic change in the score of the predicted label. To deploy punctuation-level attacks, we need to consider which positions in the input text should be modified and which punctuations should be modified. For example, if there are $k$ candidate punctuations and $n$ tokens in the input text, there will be $kn$ candidate attack texts when we insert one punctuation, $k^2 n(n+1)$ candidate attack projects when we insert two punctuations, and $k^3 n(n+1)(n+2)$ candidate attack projects when we insert three punctuations. The time complexity of multiple punctuation attacks can be reduced using a greedy algorithm or a beam search. However, even considering only a single punctuation attack is still time-consuming.

To address this problem, we propose a method called Text Position Punctuation Embedding (TPPE) to reduce search cost. The feature extraction processing of input text $\boldsymbol{x}$ is also the embedding processing of text $\boldsymbol{x}$, which is time-consuming for multiple queries. Therefore, we propose Text Position Punctuation Embedding (TPPE) to quickly and reasonably embed adversarial candidate text $\boldsymbol{x}_{adv}$. Let us illustrate it with the Insertion attack mode. When we apply Insertion attack to punctuation $p$ into the $i$-th position of the input text, we query the text function $f(\boldsymbol{x})$ or the substitute function $f_{sub}(\boldsymbol{x})$ and gain the $f_{fe}(\boldsymbol{x})$ as the embedding of text $\boldsymbol{E}_{text}$ and $f_{fe}(\boldsymbol{p})$ as the embedding of punctuation $p$. The embedding of punctuation $p$ only needs to be queried once and can be used repeatedly. As for the embedding of position, we adopt the embedding of the static position based on the calculation of sine and cosine from Transformer [40]. And we define the embedding of the adversarial candidate text $\boldsymbol{x}_{adv}$ by concatenating or adding the embedding of text $\boldsymbol{x}$, punctuation $p$ and position $i$. We can iteratively attack the input text $\boldsymbol{x}$ and quickly obtain the adversarial text $\boldsymbol{x}_{adv}$ after training the classification model from the TTPE of $\boldsymbol{x}_{adv}$ to the prediction of $\boldsymbol{x}_{adv}$ by $f(\boldsymbol{x})$. In addition, we propose a search method Text Position Punctuation Embedding and Paraphrase (TPPEP) to achieve a single-shot attack. We analyze the most information-lacking scenario for TPPEP: zero query, black-box text function, hard-label output, single punctuation limitation, and single-shot attack. To achieve the goal of zero query and gain more information, we train a substitute function $f_{sub}$ to fit the text function $f$. Then, we query the substitute function $f_{sub}$ to obtain the embedding of

text $\boldsymbol{E}_{text}$. We apply the TPPE method to obtain the embedding of the adversarial candidate text $\boldsymbol{x}_{adv}$, denoted as $\boldsymbol{E}_{\boldsymbol{x}_{adv}}$. We then transform the attacking task into the paraphrasing task. $\boldsymbol{E}_{\boldsymbol{x}_{adv}}$ and $\boldsymbol{E}_{text}$ are concatenated as input data, and the result of attack (successfully attack, donated as label 1, otherwise, donated as label 0) is the predicting label. We choose the adversarial candidate text with the highest paraphrasing score calculated by the TPPEP method and deploy the single punctuation attack in a single-shot manner accordingly.

The contributions can be summarized as follows. (1) `Punctuation-level attacks`: We first propose punctuation-level attacks, which regard the perturbations of punctuation as a systematic attack like character-level, word-level, and sentence-level attacks. We propose four primary modes of punctuation-level attacks and explain punctuation-level attacks from the perspective of optimal perturbations. (2) `TPPE`: We first propose the TPPE embedding method to decrease the search cost. We reduce the query time complexity from $\mathcal{O}\left(kn\right)$ of `Insertion`, $\mathcal{O}\left(nt\right)$ of `Displacement`, $\mathcal{O}\left(t\right)$ of `Deletion`, and $\mathcal{O}\left(kt\right)$ of `Replacement`, to $\mathcal{O}\left(1\right)$ under single punctuation attack. It can quickly and reasonably embed the adversarial candidate text $\boldsymbol{x}_{adv}$ using a single-shot query. (3) `Single-shot and Single Punctuation Attack`: To make our punctuation-level attack more imperceptible, we modify only one punctuation. Besides, we discuss single-punctuation attacks in the most challenging scenario: zero query, black-box function, hard-label output, one-punctuation limitation, and single-shot attack, which is the closest to the real-world scenarios. We correspondingly propose the TPPEP method and achieve promising experimental results.

## 2 Related Works

**Textual adversarial attack**. Text adversarial attack methods in the literature can be broadly categorized into two classes: white-box attacks and black-box attacks. In white-box attacks, some methods [23, 35, 34, 1, 6] employ FGSM [12] to identify the words or characters with the most significant impact on text-related tasks. For example, TextFool [23] introduces a gradient loss function within the model to assess the influence of specific words or characters on task performance. Suranjana *et al.* [35] consider firstly trying to remove the adverb as a word-level attack, and use a gradient loss function in the model to achieve the attack. AdvGen [1] focuses on the similarity between the original words and their substituted counterparts to target neural machine translation (NMT) models. Moreover, some methods rely on directional derivatives [9, 8]. For instance, Hot-flip [9] manipulates text function by altering characters based on directional derivative gradients. Ebrahimi *et al.* propose controlled attacks, an extension of HotFlip that enables targeted attacks [8]. In black-box attacks, specific methods aim to insert nonsensical sentences into the input text with the intention of deceiving the text model [19, 42, 4]. Notable approaches include AddSent [19] and AddSentDiverse [42]. AddSent incorporates meaningless sentences into the input text to confound comprehension system models [19]. Meanwhile, AddSentDiverse broadens the pool of adversarial candidate texts by varying the placements of inserted sentences [42]. Other strategies involve the manipulation of words and sentences [3, 29, 11, 22, 28, 2, 5, 44, 31]. For example, Textbugger [22] alters words by substituting, removing, adding, or rearranging them to deceive the text model. Furthermore, certain methods are centered on paraphrasing the input text [18, 32]. SCPNs [18] employs an encoder-decoder architecture to generate new paraphrased versions of the input text.

**Punctuation attacks**. To the best of our knowledge, there is a scarcity of research in the existing literatures on punctuation attacks. Most related studies primarily concentrate on punctuation perturbation, specifically the placement of punctuation marks within or at the end of words. This approach can be classified as a character-level attack. For example, Hosseini *et al.* insert punctuation into words to transform them into OOV words [17]. In another work by Hofer *et al.* [15], they also regard inserting a single comma as a special character-level attack, and they insert a comma at the end of the important word to translate this important word into an OOV word. In addition, a special symbol text attacks (SSTA) method is proposed in [23], which is viewed as an adjunct of TextFool [10] by adding fifty tildes behind the text.

## 3 Methodology

### 3.1 Punctuation-level Attacking

**The Proposed Punctuation-level Attacks**. We propose four modes of attacking at the punctuation-level and analyze their time complexity for traversal search. These modes are as follows:

Table 1: Punctuation-level attack modes and Complexity. $k$ is the number of candidate punctuations, $t$ is the number of punctuations in the input text and $n$ is the number of input tokens.

| mode | single punctuation | $m$ times of punctuation attacks | $m$ times of attacks by greedy algorithm |
|---|---|---|---|
| Insertion | $\mathcal{O}\left(kn\right)$ | $\mathcal{O}\left(k^m \prod_{i=0}^{m-1}(n+i)\right)$ | $\mathcal{O}(0.5k(2n+m-1))$ |
| Displacement | $\mathcal{O}\left(nt\right)$ | $\mathcal{O}\left(n^m t^m\right)$ | $\mathcal{O}\left(mnt\right)$ |
| Deletion | $\mathcal{O}(t)$ | $\mathcal{O}\left(\prod_{i=0}^{m-1}(t-i)\right)$ | $\mathcal{O}(0.5m(2t+1-m))$ |
| Replacement | $\mathcal{O}(kt)$ | $\mathcal{O}((kt)^m)$ | $\mathcal{O}(mkt)$ |

- Insertion: Punctuation $p$ is inserted into the target text to fool the text model.
- Displacement: Punctuation $p$ is moved from position $i$ to position $j$ in the target text.
- Deletion: Punctuation $p$ is removed from the target text.
- Replacement: Punctuation $p_i$ is replaced by $p_j$ in the target text.

We denote the number of candidate punctuations by $k$, the number of punctuations in the input text by $t$, and the number of input tokens by $n$. In Insertion attack, there are $n$ candidate positions, and each position can be inserted by $k$ candidate punctuations. So the complexity of Insertion by traversal search is $\mathcal{O}\left(kn\right)$. Regarding the $m$ times of Insertion attacks, one text will generate $kn$ candidate texts after a single punctuation attack, and the complexity of Insertion to each candidate is $k(n+1)$, thus the complexity of multiple Insertion attacks is $\mathcal{O}\left(k^m \prod_{i=0}^{m-1}(n+i)\right)$. If the greedy algorithm is applied for the $m$ times of Insertion attacks, the generated $kn$ candidate texts after the single punctuation attack will be selected with the greatest change in predicted score, and the number of positions is increased to $n+1$. Thus, the complexity of Insertion to this selected candidate is $\mathcal{O}(0.5k(2n+m-1))$. The complexity of Displacement by traversal search is $\mathcal{O}\left(nt\right)$ because there are $t$ candidate punctuations and each position can be displaced by $n$ candidate positions. As for the $m$ times of Displacement attacks, one text will generate $nt$ candidate texts after the single-shot attack, and the complexity of Displacement to each candidate is $nt$. So the complexity of $m$ times of Displacement attacks is $\mathcal{O}\left((nt)^m\right)$. If the greedy algorithm is applied for $m$ times of displacement attacks, there are still $t$ candidate punctuations and $n$ candidate positions. So the complexity of $m$ times of Displacement by the greedy algorithm is $\mathcal{O}(mnt)$. The complexity of Deletion by traversal search is $\mathcal{O}\left(t\right)$ because there are $t$ candidate punctuations that can be deleted. In the context of multiple occurrences of Deletion attacks, each attack leads to the removal of a candidate punctuation. Consequently, the complexity of $m$ consecutive Deletion attacks can be expressed as $\mathcal{O}\left(\prod_{i=0}^{m-1}(t-i)\right)$, where $t$ represents the total number of possible candidates. If the greedy algorithm is applied for $m$ times of Deletion attacks, due to every Deletion attack, the text will lose one punctuation, thus the complexity of $m$ times of Deletion by the greedy algorithm is $\mathcal{O}\left(0.5m(2t+1-m)\right)$. The complexity of Replacement by traversal search is $\mathcal{O}\left(kt\right)$ because there are $t$ candidate punctuations and each punctuation can be replaced by $k$ candidate punctuations. Regarding the $m$ times of replacement attacks, there are still $k$ candidate punctuations and $t$ punctuations in the input text after a single-shot attack. Thus, the complexity of $m$ times of Replacement attacks is $\mathcal{O}\left((kt)^m\right)$. If the greedy algorithm is applied for $m$ times of Deletion attacks, the complexity of $m$ times of Replacement by the greedy algorithm is $\mathcal{O}\left(mkt\right)$. All complexity is shown in Table 1.

**Single punctuation Attack**. The target text $\boldsymbol{x}$ consists of $n$ tokens, noted as $\boldsymbol{x} = [x_1, \cdots, x_n]$. According to the Universal Approximation Theorem [16], we can define a function $F(\boldsymbol{x})$ to fit the text function $f(\boldsymbol{x})$ by a continuous nonconstant, bounded, and monotonically increasing function $\phi(\cdot)$ as

$$F(\boldsymbol{x}) = \sum_{m=1}^{M} v_m \phi\left(\boldsymbol{w}_m^\top \boldsymbol{x} + b_m\right), |F(\boldsymbol{x}) - f(\boldsymbol{x})| < \epsilon, \tag{1}$$

where $v_m, b_m \in \mathbb{R}$ and $\boldsymbol{w}_m \in \mathbb{R}^D$. Universal Approximation Theorem demonstrates the fact that if our attack can result in vicious impact in $f(\boldsymbol{x})$, it will also fool the text function $F(\boldsymbol{x})$. In the

Table 2: The number of tokens affected by a single attack. $k$ is the number of candidate punctuations, $i$ is the position of attacking punctuation in the input text, and $n$ is the number of input tokens.

| Mode | Perturbed tokens | Perturbed words | Perturbed level |
|---|---|---|---|
| character-level attack | 1 | 1 | value of token |
| word-level attack | 1 | 1 | value of token |
| Insertion | $n - i + 2$ | 0 | value and position of token |
| Displacement | $j - i + 1$ | 0 | value and position of token |
| Deletion | $n - i$ | 0 | value and position of token |
| Replacement | 1 | 0 | value of token |

following, we will analyze how punctuation-level attacks influence the text function $F(\boldsymbol{x})$ based on the basic linear unit $\boldsymbol{w}_m^\top \boldsymbol{x} + b_m$ of $f(\boldsymbol{x})$ as

$$\boldsymbol{w}_m^\top \boldsymbol{x} + b_m = w_1 x_1 + w_2 x_2 + \cdots + w_m x_m + b_m. \tag{2}$$

We compare character-level, word-level, and punctuation-level attacks under the constraint of a single-element attack, i.e., perturbing one character, one word, or one punctuation. In character-level attacks, a single-character attack impacts only one token $x_i$, making it OOV. In many text functions, the OOV token is embedded as an identical token $x_{oov}$. In this scenery, $\boldsymbol{w}_m^\top \boldsymbol{x} + b_m$ can be written as

$$\boldsymbol{w}_m^\top \boldsymbol{x} + b_m = w_1 x_1 + w_2 x_2 + \cdots + w_i x_{oov} + \cdots + w_m x_m + b_m. \tag{3}$$

Similar to character-level attacks, a single-word attack impacts only one token $x_i$. Since there are few candidate tokens $x_{can}$ that are semantically similar to the token $x_i$ when considering the similarity of input text with and without a word-level attack, $\boldsymbol{w}_m^\top \boldsymbol{x} + b_m$ can be rewritten as

$$w_1 x_1 + w_2 x_2 + \cdots + w_i x_{can} + \cdots + w_m x_m + b_m. \tag{4}$$

Regarding the proposed punctuation attacks, we discuss single punctuation attacks in terms of Insertion, Displacement, Deletion, and Replacement respectively. Taking Insertion as an example, when we insert punctuation $p$ into position $i$ of the target text, $\boldsymbol{w}_m^\top \boldsymbol{x} + b_m$ will change to

$$w_1 x_1 + w_2 x_2 + \cdots + w_i p + w_{i+1} x_i + \cdots + + w_m x_{m-1} + w_{m+1} x_m + b_m. \tag{5}$$

According to Eq. (3) and (4), the character-level and word-level attacks only perturb one token and one word by the single punctuation attack. According to Eq. (5), the single Insertion attack does not perturb any words but perturbs the $i$-th token by substituting $x_i$ with punctuation $p$ and all tokens after the $(i-1)$-th position of the input text by moving them one bit backward, which totally perturbs $n - i + 2$ tokens. The result of Displacement, Deletion, and Replacement is presented in Table 2. According to Table 2, the punctuation-level attack can impact more tokens, indicating that it can cause greater perturbation than character- and word-level attacks. If we insert, displace, or delete punctuation in the appropriate position, it will result in a more significant impact on the basic linear $\boldsymbol{w}_m^\top \boldsymbol{x} + b_m$ of $f(\boldsymbol{x})$ compared to character-level and word-level attacks. Meanwhile, a single punctuation attack may apply a more vicious impact to $f(\boldsymbol{x})$ and $F(\boldsymbol{x})$. Compared to words, punctuation is thinner, smaller, and more imperceptible. Due to the nature that punctuation-level attack does not perturb any word, semantically, the adversarial text is more imperceptible for human beings. Furthermore, the Insertion, Displacement, Deletion, and Replacement of the punctuation produces less impact on the semantic information by human beings understanding.

**The Ultimate Effectiveness of Punctuation-level Attacks**. According to Eq. (3), (4), (5), character- and word-level attacks under a single-element limitation may focus on perturbing token values. However, Insertion, Displacement, and Replacement attacks can not only perturb token values but also token positions which may result in a vicious impact on $f(\boldsymbol{x})$. This position perturbation almost has no impact on semantic information for human beings.

We discuss why multiple punctuation-level attacks can fool the text model by taking Insertion attacks as an example. In the text model, there exists the maximum length of the input text, and we denote it as $N$. If the length of the input text is greater than $N$, the text model $f$ considers only the first $N$ tokens. If we apply $N$ Insertion attacks, there will be $k^N$ candidate texts. The probability of fooling the text model is equal to 1 minus the probability of all $k^N$ candidate texts failing to fool

Table 3: Text embedding in punctuation-level attack.

| Mode | Concat | mathematical operation |
|---|---|---|
| Insertion | $\boldsymbol{E}_{all}$ | $\boldsymbol{E}_{text} + \boldsymbol{E}_{pos} + \boldsymbol{E}_{punc}$ |
| Displacement | $\boldsymbol{E}_{all}$ | $\boldsymbol{E}_{text} - \boldsymbol{E}_{pos}^i + \boldsymbol{E}_{pos}^j + \boldsymbol{E}_{punc}$ |
| Deletion | $\boldsymbol{E}_{all}$ | $\boldsymbol{E}_{text} - \boldsymbol{E}_{pos}^i - \boldsymbol{E}_{punc}$ |
| Replacement | $\boldsymbol{E}_{all}$ | $\boldsymbol{E}_{text} + \boldsymbol{E}_{pos} - \boldsymbol{E}_{punc}^i + \boldsymbol{E}_{punc}^j$ |

the text model. We denote the probability of fooling the text model as $p_{fool}$ and the probability of failing to fool the text model by the $i$-th candidate text as $p_i, i \in [1, K^N]$. And we denote the maximum value of $p_i$ as $p_{max}$. We make an assumption about $p_{max}$ here. Due to the fact that $p_i$ is the probability of the $i$-th candidate text before querying the text model $f$, and the attacking result is unknown, we have $0 < p_{max} < 1$. Correspondingly, we have

$$p_{fool} = 1 - \prod_{i=1}^{k^N} P_i \geq 1 - p_{\max}^{k^N}. \tag{6}$$

According to Eq. (6), $1 - p_{max}^{k^N}$ will approach 1 as $k$ increases. As a probability, we have $1 - p_{max}^{k^N} \leq p_{\text{fool}} \leq 1$. Consequently, $p_{fool}$ will tend to 1 when $k$ increases, which means multiple Insertion attacks can ultimately fool the text model with $k$ increasing.

## 3.2 Embedding and Search Methods

According to Table 1, punctuation-level attacks require considerable cost to deploy. Even considering only the single punctuation attack, it is still time-consuming. Taking the mode of Insertion as an example, its original time complexity is $\mathcal{O}(kn)$. Therefore, it is essential to decrease the search cost. If we have reduced the search cost of a single punctuation attack, the time complexity of multiple-punctuation attacks can be reduced by using either a greedy algorithm or a beam search. The $n$ is determined by the target text $\boldsymbol{x}$, which is not feasible to optimize. As $k$ increases, the search space expands, resulting in a higher fooling rate of Insertion attacking. Therefore, to achieve a higher fooling rate, we should not reduce the value of $k$.

As we cannot decrease $k$ and $n$, we focus on a specific process: text function $f(\boldsymbol{x})$. For example, in a text classification task, function $f(\boldsymbol{x}) = \text{softmax } f_{fe}(\boldsymbol{x})$, where the processing of feature extraction is denoted by the function $f_{fe}$ and softmax is used as the final classification layer. The classification cost $t_0$ can be divided into two parts: the feature extraction consumption $t_1$ and the final classification consumption $t_2$. The classification time $t_2$ is transient compared to $t_1$ and therefore can be ignored. Consequently, we have $t_0 \approx t_1$.

Embedding the adversarial candidate text $\boldsymbol{x}_{adv}$ perturbed by various configurations requires multiple queries, and thus it is time-consuming. Therefore, we propose TPPE to quickly and reasonably embed the adversarial candidate text $\boldsymbol{x}_{adv}$. According to Eq. (5), a single punctuation attack consists of three components: input text $\boldsymbol{x}$, perturbed position, and candidate punctuation. To reduce the cost of embedding processing for adversarial text $\boldsymbol{x}_{adv}$, we refine its embedding process by combining the embeddings of text, position, and punctuation. Again, we take the Insertion attack mode as an example. When we insert punctuations $p$ into the $i$-th position of the target text, we query text function $f(\boldsymbol{x})$ (white-box) or substitute function $f_{sub}(\boldsymbol{x})$ (black-box), obtaining $\boldsymbol{E}_{text}$ as the embedding of text and $f_{fe}(\boldsymbol{p})$ as the embedding of punctuation $p$. The embedding of punctuation $p$ only needs to be queried once. For the embedding of the position, we adopt the embedding of the static position based on the calculation of sine and cosine from the Transformer [40], present in Eq. (7). The variable $pos$ represents the position and the variable $i$ represents the dimension.

$$PE_{(pos,2i)} = \sin\left(pos/10000^{2i/d_{\text{model}}}\right), PE_{(pos,2i+1)} = \cos\left(pos/10000^{2i/d_{\text{model}}}\right). \tag{7}$$

After calculating embeddings of input text, punctuation, and position, we calculate candidate attacking text embeddings using two steps: 1) concatenating three embeddings as candidate attacking text embedding $\boldsymbol{E}_{all} = Concat(\boldsymbol{E}_{text}, \boldsymbol{E}_{pos}, \boldsymbol{E}_{punc})$ and, 2) calculating candidate attacking text embedding using mathematical operations presented in Table 3.

Table 4: The quantitative results of single punctuation attack in the T2I task. "Ori-text0" indicates the original text "a professional photograph of an astronaut riding a triceratops", and "Ori-text1" indicates the original text "a corgi is playing piano, oil on canvas" .

| dataset | pokemon-blip-captions | | | | Ori-image | Adv-image | Ori-text | Adv-text |
|---------|------|-------|------|----------|-----------|-----------|----------|----------|
| | all | train | test | | | | | |
| Ori-text | 0.3273 | 0.3272 | 0.3278 | Ori-text0 | 0.3281 | 0.2484 | 1 | 0.9782 |
| Adv-text | 0.2591 | 0.2586 | 0.2610 | Ori-text1 | 0.4040 | 0.3468 | 1 | 0.9843 |

The `Insertion` mode inserts punctuation $p$ into the $i$-th position of input text, so we regard $\boldsymbol{E}_{text} + \boldsymbol{E}_{pos} + \boldsymbol{E}_{punc}$ as the embedding of $\boldsymbol{x}_{adv}$. The `Displacement` mode displaces punctuation $p$ from position $i$ to position $j$ in the input text, so we regard $\boldsymbol{E}_{text} - \boldsymbol{E}_{pos}^i + \boldsymbol{E}_{pos}^j + \boldsymbol{E}_{punc}$ as the embedding of $\boldsymbol{x}_{adv}$. The `Deletion` mode deletes punctuation $p$ from the $i$ th position of the input text, so we compute $\boldsymbol{E}_{text} + \boldsymbol{E}_{pos} - \boldsymbol{E}_{punc}^i + \boldsymbol{E}_{punc}^j$ as the embedding of $\boldsymbol{x}_{adv}$. The `Replacement` mode replaces punctuation $p_i$ with $p_j$ in the input text, so we take $\boldsymbol{E}_{text} + \boldsymbol{E}_{pos} - \boldsymbol{E}_{punc}^i + \boldsymbol{E}_{punc}^j$. TPPE requires only single-shot embedding of the input text $\boldsymbol{x}$, which means we decrease the query time complexity from $\mathcal{O}(kn)$ of `Insertion`, $\mathcal{O}(nt)$ of `Displacement`, $\mathcal{O}(t)$ of `Deletion`, and $\mathcal{O}(kt)$ of `Replacement`, to $\mathcal{O}(1)$ under single punctuation attack.

In our study, we assume the worst-case of applying punctuation-level attacks: the victim model is a black-box model where only prediction labels are available instead of scores from the function. In this scenery, we train a substitute function $f_{sub}$ [30] to transform the black-box scenario into a white-box scenario by collecting part of training datasets $\boldsymbol{X}$. Specifically, we query the text function and derive the prediction labels $y_{pre}$, then train the substitute function $f_{sub}$ using the input text and the label $y_{pre}$ as paired data. After training $f_{sub}$, we transform the black-box scenario into a white-box scenario.

Directly querying $f_{sub}$ to determine which punctuation should be used is also time-consuming for multiple queries. We iteratively attack the input text $\boldsymbol{x}$ and quickly gain the adversarial text $\boldsymbol{x}_{adv}$ after training the classification model from the TPPE of $\boldsymbol{x}_{adv}$ to the label of $\boldsymbol{x}_{adv}$ by $f(\boldsymbol{x})$. Additionally, we propose a search method called TPPEP to achieve a single-shot attack. We analyze the worst-case scenario for TPPEP: zero query, black-box function, hard-label output, single punctuation limitation, and single-shot attack. The specific pseudo code of TPPE and TPPEP are presented in Alg. 1 and Alg. 2 in the Appendix.

To achieve the goal of zero query and gain more information, we train a substitute function $f_{sub}$ to fit the text function $f$. Then, we query the substitute function $f_{sub}$ to gain the embedding of text $\boldsymbol{E}_{text}$. We apply the TPPE method to gain the embedding of adversarial candidate text $\boldsymbol{x}_{adv}$, denoted as $\boldsymbol{E}_{\boldsymbol{x}_{adv}}$. We transform the attacking task into a paraphrasing task. Specifically, $\boldsymbol{E}_{\boldsymbol{x}_{adv}}$ and $\boldsymbol{E}_{text}$ are concatenated as input data, and the result of the attack (the successful attack denotes label 1; else the label 0) is the predicting label. The adversarial candidate text with the highest paraphrasing score calculated by the TPPEP method is chosen to deploy the attack.

### 3.3  Analysis of TPPEP

In this section, we discuss a specific and practical form of classification function that can deceive the text model. We assume that there is a function $f(\boldsymbol{x})$ in hypothesis space $\mathcal{F}$. Since $f(\boldsymbol{x}) = \text{softmax}(f_{fe}(\boldsymbol{x}))$, we hypothesize that there exists a function $f_1$ that fits the function softmax and satisfies $|f_1(\boldsymbol{x}) - \widehat{y}_{\boldsymbol{x}}| < \epsilon_1$. A simple functional form of $f_1$ is a function of $\arg\max \text{softmax}$. Furthermore, we propose the hypothesis that there exists a function $f_2$, which can predict $\widehat{y}_{\boldsymbol{x}_{adv}}$ from $TPPE(\boldsymbol{x}_{adv})$ and satisfy $|f_2(\boldsymbol{x}_{adv}) - \widehat{y}_{\boldsymbol{x}_{adv}}| < \epsilon_2$. We also define a function $f_3$,

$f_3(\boldsymbol{x}_{adv}, \boldsymbol{x}) = \begin{cases} 1 & \text{if } |f_2(\boldsymbol{x}_{adv}) - f_1(\boldsymbol{x})| \geq 1 \\ 0 & \text{otherwise} \end{cases}$ . This means we can predict the result of an attack

by $|f_2(\boldsymbol{x}) - f_1(\boldsymbol{x})|$. We calculate the absolute difference between $f_2(\boldsymbol{x}_{adv})$ and $f_1(\boldsymbol{x})$ and compare the absolute difference between $|f_2(\boldsymbol{x}_{adv}) - \widehat{y}_{\boldsymbol{x}_{adv}}|$ and $|\widehat{y}_{\boldsymbol{x}_{adv}} - \widehat{y}_{\boldsymbol{x}}|$ in Eq. (8) as

$$
\begin{aligned}
|f_2(\boldsymbol{x}_{adv}) - f_1(\boldsymbol{x})| &= |f_2(\boldsymbol{x}_{adv}) - \widehat{y}_{\boldsymbol{x}_{adv}} + \widehat{y}_{\boldsymbol{x}_{adv}} - f_1(\boldsymbol{x}) + \widehat{y}_{\boldsymbol{x}} - \widehat{y}_{\boldsymbol{x}}| \\
&= |[f_2(\boldsymbol{x}_{adv}) - \widehat{y}_{\boldsymbol{x}_{adv}}] + [\widehat{y}_{\boldsymbol{x}_{adv}} - \widehat{y}_{\boldsymbol{x}}] + [\widehat{y}_{\boldsymbol{x}} - f_1(\boldsymbol{x})]| \\
&\leq \varepsilon_1 + \varepsilon_2 + |\widehat{y}_{\boldsymbol{x}_{adv}} - \widehat{y}_{\boldsymbol{x}}|.
\end{aligned} \tag{8}
$$

According to Eq. (8), when $\widehat{y}_{\boldsymbol{x}_{adv}} = \widehat{y}_{\boldsymbol{x}}$, the actual label of TPPEP is 0 and $\mid f_2(\boldsymbol{x}_{adv}) - f_1((\boldsymbol{x})) \leq \varepsilon_1 + \varepsilon_2$. Since $\varepsilon_1$ and $\varepsilon_2$ are sufficiently small, $f_3(f_2(\boldsymbol{x}_{adv}), f_1(\boldsymbol{x}))$ will be 0, allowing us to classify them correctly. As $f_1(x)$ and $f_2(x)$ fit well $\widehat{y}_{\boldsymbol{x}_{adv}}$ and $\widehat{y}_{\boldsymbol{x}}$, when $\widehat{y}_{\boldsymbol{x}_{adv}}$ and $\widehat{y}_{\boldsymbol{x}}$ are not equal, $f_3(f_2(\boldsymbol{x}_{adv}), f_1(\boldsymbol{x}))$ will be 1. Thus, we have found a specific fooling function $f_3(f_2(\boldsymbol{x}_{adv}), f_1(\boldsymbol{x}))$, which implies that there is at least one fooling function in the hypothesis space $\mathcal{F}$. Moreover, we discuss that $f_3(f_2(\boldsymbol{x}_{adv}), f_1(\boldsymbol{x}))$ can be learned based on the Universal Approximation Theorem [16]. According to Eq. (1), we can train $F_1$ and $F_2$ to fit $f_1$ and $f_2$, which satisfy $|F_1(\boldsymbol{x}) - f_1(\boldsymbol{x})| < \epsilon_3$ and $|F_2(\boldsymbol{x}_{adv}) - f_2(\boldsymbol{x}_{adv})| < \epsilon_4$. We analyze the difference between $|f_2(\boldsymbol{x}_{adv}) - f_1(\boldsymbol{x})|$ and $|F_2(\boldsymbol{x}_{adv}) - F_1(\boldsymbol{x})|$ in Eq. (9). If the difference between them is less than 1, then $f_3(F_2(\boldsymbol{x}_{adv}), F_1(\boldsymbol{x}))$ and $f_3(f_2(\boldsymbol{x}_{adv}), f_1(\boldsymbol{x}))$ will predict the same label.

$$|[F_2(\boldsymbol{x}_{adv}) - F_1(\boldsymbol{x})] - [f_2(\boldsymbol{x}_{adv}) - f_1(\boldsymbol{x})]| = |[F_1(\boldsymbol{x}) - f_1(\boldsymbol{x})] + [f_2(\boldsymbol{x}_{adv}) - F_2(\boldsymbol{x}_{adv})])| \\ \leq |F_1(\boldsymbol{x}) - f_1(\boldsymbol{x})| + |F_2(\boldsymbol{x}_{adv}) - f_2(\boldsymbol{x}_{adv})| \leq \varepsilon_3 + \varepsilon_4. \tag{9}$$

According to Eq. (9), we compare the difference between $|f_2(\boldsymbol{x}_{adv}) - f_1(\boldsymbol{x})|$ and $|F_2(\boldsymbol{x}_{adv}) - F_1(\boldsymbol{x})|$. As $\varepsilon_3$ and $\varepsilon_4$ are sufficiently small, $|F_2(\boldsymbol{x}) - F_1(\boldsymbol{x})|$ will approximate $|f_2(\boldsymbol{x}) - f_1(\boldsymbol{x})|$, suggesting that we can fit $|\widehat{y}_{\boldsymbol{x}_{adv}} - \widehat{y}_{\boldsymbol{x}}|$ by $|f_2(\boldsymbol{x}) - f_1(\boldsymbol{x})|$. Based on the above analysis, we prove that a feasible solution exists in the hypothesis space and can be trained.

## 4  Experiment

### 4.1  Experimental Setting

**Tasks and Datasets**. We have implemented punctuation-level attacks and TPPEP in the context of three distinct text-to-label tasks, utilizing three diverse datasets. Furthermore, we have reported the outcomes of these experiments concerning text summarization, semantic-similarity-scoring, and text-to-image tasks, focusing particularly on the single-shot punctuation attack. The tasks that underwent single-shot attacks and TPPEP perturbations encompassed text classification (TC), paraphrasing, and natural language inference (NLI) tasks. The TC task takes a text as input, and the text model generates a predicted label for the input text. For our TC task, we have opted to utilize the CoLA dataset [43]. CoLA is the dataset of a binary classification task that aims to predict whether the input sentence is "correct" from a linguistic perspective. As for the paraphrase task, the input consists of a pair of texts, and the objective of the text model is to ascertain the semantic similarity between the two texts. Our choice for the paraphrasing task is the QQP dataset [21]. Furthermore, the NLI task involves predicting the relationship between pairs of input sentences and we employ the Wanli dataset [25].

**Experiment Setup**. We have applied both punctuation-level attacks and TPPEP to target pre-trained SOTA models, including ELECTRA [7], XLMR [37], DistilBERT1 [36], RoBERTa [27], and DeBERTa [14]. In numerous real-world text API application scenarios, the API solely provides the predicted label for the input text. Therefore, our experiments are tailored for black-box attacks, where the text model exclusively generates categorical labels in response to queries. The conventional perturbation budget of character-/word-/sentence-level attacks is typically assessed based on the number of altered words. In contrast, our punctuation-level attacks perturb zero word, and we define the perturbation budget as the number of perturbed tokens. In the experiment, we set the strictest perturbation limit, specifically focusing on the perturbation of a single punctuation. Thus, we discuss the results in the case of a single punctuation attack. In addition, to make our TPPEP work in hard-label attacks, we first train a substitute function $f_{sub}$ [30] by querying the text function and deriving the prediction labels $y_{pre}$. Then, we apply the proposed TPPEP method to search for the most vulnerable candidate text.

### 4.2  Experimental Results

**Text classification task, paraphrase task, and natural language inference task**. The result of the TC task is presented in Table 5. Our punctuation-level attack shows encouraging results on ELECTRA [7] model and Cola dataset with $90.80\%$ and $93.67\%$ fool rates respectively by single-shot `Insertion` attack. On all three datasets, `Insertion` consistently obtains the highest fool rate due to the largest search space compared to other attacks such as `Deletion`, which has the smallest search space and thus obtains the lowest fool rate. The fool rate of `Displacement` and `Replacement` depends more on the specific datasets and text models. The results of the traversal

Table 5: The results on Cola, QQP, and Wanli datasets. Top-1 is the fool rate of single-shot and single punctuation attack. The Top-3 and Top-5 are the fool rates of three and five single punctuation attacking candidate texts. Traversal is the fool rate of a single punctuation attack by traversal, which is the maximum value of a single punctuation attack. $p_1$ is the ratio of TOP-1 to traversal. Average search space (ASP) is the mean of candidate texts.

| Cola | ELECTRA [7] | | | | | XLMR [37] | | | | | |
|---|---|---|---|---|---|---|---|---|---|---|---|
| mode | Top-1 | Top-3 | Top-5 | Traversal | $p_1$ | Top-1 | Top-3 | Top-5 | Traversal | $p_1$ | ASP |
| Insertion | 67.40% | 73.06% | 73.83% | 90.80% | 74.23% | 28.76% | 52.64% | 63.57% | 93.67% | 30.70% | 362.62 |
| Displacement | 36.05% | 66.35% | 73.44% | 80.44% | 44.82% | 43.05% | 60.12% | 76.03% | 80.73% | 53.33% | 11.59 |
| Deletion | 5.18% | 5.85% | 5.94% | 5.94% | 87.21% | 4.89% | 5.85% | 5.85% | 5.85% | 83.59% | 1.15 |
| Replacement | 24.64% | 36.82% | 44.77% | 74.59% | 33.03% | 6.62% | 9.88% | 12.37% | 20.23% | 32.72% | 41.49 |
| QQP | DistilBERT1 [36] | | | | | DistilBERT2 [36] | | | | | |
| Insertion | 14.72% | 18.76% | 22.68% | 47.18% | 31.20% | 8.67% | 10.43% | 11.73% | 48.23% | 17.98% | 957.72 |
| Displacement | 8.52% | 15.05% | 18.86% | 26.78% | 31.81% | 7.21% | 12.43% | 15.57% | 23.44% | 30.76% | 36.57 |
| Deletion | 3.94% | 5.93% | 6.02% | 6.03% | 65.34% | 5.06% | 6.86% | 6.95% | 6.96% | 72.70% | 2.53 |
| Replacement | 7.59% | 10.04% | 12.18% | 19.70% | 38.53% | 16.70% | 20.97% | 22.65% | 29.65% | 56.32% | 90.91 |
| Wanli | RoBERTa [27] | | | | | DeBERTa [14] | | | | | |
| Insertion | 8.44% | 19.22% | 26.20% | 66.74% | 12.65% | 15.28% | 29.20% | 37.40% | 80.14% | 19.07% | 1161.12 |
| Displacement | 5.12% | 9.14% | 12.26% | 26.14% | 19.59% | 10.28% | 16.60% | 20.34% | 38.40% | 26.77% | 53.94 |
| Deletion | 3.22% | 5.84% | 6.14% | 6.16% | 52.27% | 5.74% | 8.58% | 8.96% | 8.98% | 63.92% | 2.94 |
| Replacement | 8.48% | 15.96% | 19.80% | 45.82% | 18.51% | 6.92% | 13.08% | 16.88% | 54.76% | 12.64% | 105.88 |

Table 6: The results of the semantic-similarity-scoring task

| | Sentence-BERT | | Distilbert | |
|---|---|---|---|---|
| STS-B | Pearson | Spearman | Pearson | Spearman |
| Without Attack | 0.7990 | 0.6988 | 0.8056 | 0.7257 |
| TOP-1 | 0.7874 | 0.6862 | 0.7902 | 0.7035 |
| TOP-3 | 0.7760 | 0.6738 | 0.7759 | 0.6990 |
| TOP-5 | 0.7654 | 0.6626 | 0.7649 | 0.6745 |
| Traversal | 0.6992 | 0.5832 | 0.6994 | 0.6048 |

search imply that a single punctuation attack can fool those text models. The results of Top-1 present the result of single-shot and single punctuation attack. Our experiments show that single-shot and single punctuation attacks are more effective for datasets with small average search space (ASP) such as the CoLA dataset. In the case of ELECTRA model and CoLA dataset, single-shot and single punctuation attack obtains 67.40%, 36.05%, 5.18%, and 24.64% respective fool rate by `Insertion`, `Displacement`, `Deletion`, and `Replacement` attack. The fool rate of the single-shot and single punctuation attack decreases with increasing search space but still achieves a fool rate of 8.44% when the ASP is 1161.12 in the Wanli dataset.

**Semantic-similarity-scoring task, summarization task, and text-to-image task**. In addition, we have expanded the scope of single punctuation attacks to encompass semantic-similarity-scoring (sss), summarization, and text-to-image tasks (T2I). Datasets from STS-B [26] and gigaword [20] datasets were chosen for the sss and summarization tasks. Results of single punctuation attacks are tabulated in Tables 6 and 7. In the sss task, the `Insertion` punctuation attack leads to reductions of Pearson and Spearman correlations for Sentence-BERT and Distilbert, resulting in reductions of 0.0998, 0.1156, 0.1062, and 0.1209, respectively. For the summarization task, we utilize the ROUGE-1 metric as the dependent variable, denoted as $y$. Then we formulate a predictive model to estimate the ROUGE-1 score for the test set. Subsequently, we select the candidate attack text with the lowest predicted ROUGE-1 value. The `Insertion` punctuation attack decreases the ROUGE-1 score [24] of gigaword [20] by 6.47. In the sss task, $y$ represents the semantic similarity between the text before and after the attack. To model this, we have developed a predictive model to estimate semantic similarity values for the test set. This significant decline demonstrates the susceptibility of sss and summarization models to single punctuation attack.

In the T2I task, we determine the CLIP score denoted as $y$ and select the candidate attack text with the lowest predicted CLIP score. Stable Diffusion V2 [33] is selected as the victim model, and the results are illustrated in Fig. 2. When we input the sentence "a professional photograph of an astronaut riding a triceratops", the Stable Diffusion V2 generates the "correct" image containing both the astronaut and triceratops. However, after inserting a period into the input text, Stable Diffusion V2 generates an image without triceratops. This dramatic change also happens in another input sentence. Furthermore, we have expanded the section dedicated to quantitative experiments in T2I applications,

Table 7: The results of text to image and summarization task

| Task | Metric | Without Attack | TOP-1 | TOP-3 | TOP-5 | Traversal |
|------|--------|----------------|-------|-------|-------|-----------|
| Text to image | CLIP score | 0.3278 | 0.3176 | 0.3069 | 0.3022 | 0.2610 |
| Summarization | ROUGE-1 | 11.69 | 10.91 | 9.65 | 9.11 | 5.22 |

**Ori-output**      **Adv-output**              **Ori-output**      **Adv-output**

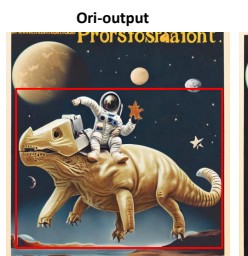 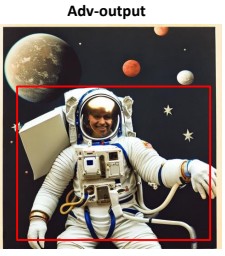 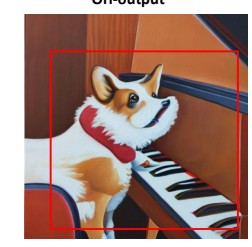 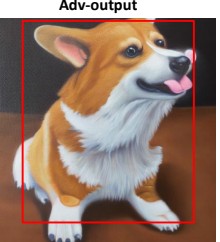

**Ori-text: a professional photograph of an astronaut riding a triceratops**
**Adv-text: a professional photograph of an astronaut. riding a triceratops**

**Ori-text: a corgi is playing piano, oil on canvas**
**Adv-text: a corgi is playing, piano, oil on canvas**

Figure 2: The results of single punctuation attack in the text-to-image task. "Ori-text" indicates the original text and "Adv-text" indicates the adversarial text under a single punctuation attack. "Ori-output" and and "Adv-output" indicate the images generated by the original text and the adversarial text respectively. The results reveal that the text-to-image model is fooled by inserting a comma or period into the input text.

utilizing the CLIP score as our evaluation metric. The experimental results are displayed in Table 4, with "ori-text1" denoting the sentence "a corgi is playing the piano, oil on canvas." By inserting a comma as a punctuation-level attack, "adv-text1" is derived from "ori-text1". Significantly, despite a high semantic similarity of 0.9843 between "ori-text1" and "adv-text1", the CLIP score decreases from 0.4040 to 0.3468 after the attack. Furthermore, to evaluate the resilience of the Stable Diffusion model against punctuation-level attacks, we utilize the "pokemon-blip-captions" dataset. Following the punctuation `insertion` attacks, the overall CLIP score decreases from $0.3273$ to $0.2591$. This reduction is consistently observed in both the training and testing subsets, highlighting the model's susceptibility. In the present the results of our endeavor to extend punctuation-level attacks, TPPE, and TPPEP algorithms to encompass three distinct tasks: summarization, semantic similarity scoring, and text-to-image generation show the effectiveness of TPPE and TPPEP.

## 5 Conclusion

In this work, we propose a punctuation-level attack based entirely on inserting, displacing, deleting, and replacing punctuation to fool text models. Our punctuation-level attack is more imperceptible to human beings and has less semantic impact compared to character-/word-/sentence-level attack. We introduce TPPE and TPPEP methods to reduce the search cost and achieve the time complexity of the query as $\mathcal{O}(1)$. Furthermore, we discuss the ultimate effectiveness of punctuation-level attacks and present the analysis of the TPPEP method. Experimental studies on representative datasets of SOTA victim models demonstrate the effectiveness of our proposed methods.

## Acknowledgment

This work was supported in part by the National Key R&D Program of China (Grant No. 2022ZD0118100), in part by the National Natural Science Foundation of China (Grant No. 62372480), in part by the Guangdong Basic and Applied Basic Research Foundation (No. 2023A1515012839), in part by ARC-Discovery grant projects (DP 190102261 and DP220100800), and in part by the Shenzhen Science and Technology Program (No. JSGG20220831093004008), and sponsored by CCF-Tencent Rhino-Bird Open Research Fund (No. CCF-Tencent RAGR20230118).

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
