# A Appendix

## A.1 TPPE Method

We present the pseudo code for TPPE in this paper, using the `Insertion` mode as an example.

---
**Algorithm 1** TPPE Embedding Method of `Insertion`

---
**Input:** The input text $\boldsymbol{x}$, the number of tokens $n$, the candidate punctuations $p_i$, the feature extraction function $f_{fe}(\boldsymbol{x})$
**Output:** the embedding of adversarial candidate text $\boldsymbol{x}_{adv}$
    **for** $i = 1$ to $n$ **do**
        $\boldsymbol{E}_{pos}^{i} = PE(i)$
    **end for**
    **for** $i = 1$ to $k$ **do**
        $\boldsymbol{E}_{punc}^{i} = f_{fe}(\boldsymbol{p_i})$
    **end for**
    $\boldsymbol{E}_{text} = f_{fe}(\boldsymbol{x})$
    **for** $i = 1$ to $n$ **do**
        **for** $j = 1$ to $k$ **do**
            $\boldsymbol{E}_{\boldsymbol{x}_{adv}^{ij}} = \boldsymbol{E}_{text} + \boldsymbol{E}_{posi} + \boldsymbol{E}_{punc}$
        **end for**
    **end for**
    $\boldsymbol{E}_{\boldsymbol{x}_{adv}} = \left[ \boldsymbol{E}_{\boldsymbol{x}_{adv}^{11}}, \boldsymbol{E}_{\boldsymbol{x}_{adv}^{12}}, \ldots, \boldsymbol{E}_{\boldsymbol{x}_{adv}^{ik}}, \boldsymbol{E}_{\boldsymbol{x}_{adv}^{21}}, \ldots, \boldsymbol{E}_{\boldsymbol{x}_{adv}^{nk}} \right]$
    **return** $\boldsymbol{E}_{\boldsymbol{x}_{adv}}$

---

According to Alg. 1, we reduce the query time complexity from $\mathcal{O}(kn)$ of `Insertion` to $\mathcal{O}(1)$ by using the TPPE method. At the same time, we can infer that the time complexity of other attack modes also becomes $\mathcal{O}(1)$.

## A.2 Substitute Model

In our study, we assume the worst-case scenario of applying punctuation-level attacks. The victim model is a black-box model where only prediction labels are available instead of function scores. In this scenario, we train a substitute function $f_{sub}$ to transform the black-box scenario into a white-box scenario by collecting the training datasets $\boldsymbol{X}$. Specifically, we query the text function and derive the prediction labels $y_{pre}$. Then, we train the substitute function $f_{sub}$ using the input text and label $y_{pre}$ as paired data. After training $f_{sub}$, we transform the black-box scenario into a white-box scenario.

**Substitute Datasets**. We denote the training datasets as $\boldsymbol{X}$ and obtain the adversary labels of $\boldsymbol{X}$ after querying the text function $f$. The adversary labels are denoted as $y_{pre}$. In the replacement training process, we use $\boldsymbol{X}$ as input data and the loss of cross-entropy as the loss function for the replacement function $f_{sub}$.

**Substitute Architecture**. In the substitute model training process, we embed the input text using the Bert pre-trained model. Two fully connected layers are adopted after the embedding layer, and a Softmax layer is adopted to predict the label of the input text.

**Substitute Training Algorithm**. We train the substitute function $f_{sub}$ for 10 epochs with a learning rate of 0.00002, a batch size of 100, and stop training when the loss of the validation data is less than 0.001. The selected substitute architecture is adopted to train the substitute model of the text model $f$.

## A.3 TPPEP Method

Directly querying $f_{sub}$ to determine which punctuation should be deployed is also time-consuming due to multiple queries. Instead, we can iteratively attack the input text $\boldsymbol{x}$ and quickly gain the adversarial text $\boldsymbol{x}_{adv}$ after training the classification model from the TPPE of $\boldsymbol{x}_{adv}$ to the prediction of $\boldsymbol{x}_{adv}$ by $f(\boldsymbol{x})$. We also propose a search method called Text Position Punctuation Embedding and Paraphrase (TPPEP) to achieve a single-shot attack. We analyze the worst-case scenario for TPPEP:

Table 8: The results of the effect of position on the fooling rate

|  | preceding | middle | subsequent |
|---|---|---|---|
| Insertion | 25.48% | 38.81% | 35.72% |
| Displacement | 41.70% | 42.00% | 16.31% |
| Replacement | 17.08% | 38.91% | 44.01% |
| Deletion | 17.48% | 39.89% | 42.62% |

Table 9: The results of multiple attacks

| Dataset | model | mode | Top-10 | Top-20 | Top-30 |
|---|---|---|---|---|---|
| Cola | ELECTRA | Insertion | 74.59% | 75.17% | 76.51% |
|  |  | Displacement | 77.89% | 78.93% | 79.41% |
|  |  | Replacement | 50.60% | 59.31% | 63.92% |
|  |  | Deletion | 5.94% | 5.94% | 5.94% |
| Cola | XLMR | Insertion | 76.51% | 83.41% | 85.81% |
|  |  | Displacement | 80.29% | 80.73% | 80.73% |
|  |  | Replacement | 15.20% | 18.14% | 19.61% |
|  |  | Deletion | 5.85% | 5.85% | 5.85% |
| QQP | DistillBERT1 | Insertion | 26.86% | 30.61% | 32.68% |
|  |  | Displacement | 23.46% | 26.16% | 26.67% |
|  |  | Replacement | 14.77% | 16.87% | 17.93% |
|  |  | Deletion | 6.03% | 6.03% | 6.03% |
| QQP | DistillBERT2 | Insertion | 29.21% | 31.81% | 35.55% |
|  |  | Displacement | 19.88% | 22.50% | 23.09% |
|  |  | Replacement | 24.46% | 26.48% | 27.54% |
|  |  | Deletion | 6.96% | 6.96% | 6.96% |
| Wanli | RoBERTa | Insertion | 37.02% | 45.36% | 49.60% |
|  |  | Displacement | 15.96% | 20.41% | 23.07% |
|  |  | Replacement | 25.58% | 33.11% | 36.89% |
|  |  | Deletion | 6.16% | 6.16% | 6.16% |
| Wanli | DeBERTa | Insertion | 44.70% | 53.00% | 56.00% |
|  |  | Displacement | 26.74% | 32.65% | 35.41% |
|  |  | Replacement | 22.51% | 29.34% | 33.97% |
|  |  | Deletion | 8.98% | 8.98% | 8.98% |

zero query, black-box function, hard-label output, single-punctuation limitation, and single-shot attack. We describe the TPPEP method as being decomposed into two parts: training and searching.

**TPPEP Training Algorithm**. To achieve the goal of zero query, the substitute function $f_{sub}$ is trained to fit the text function $f$. We query $f_{sub}$ to obtain the embedding of the text $\boldsymbol{E}_{text}$ and apply the TPPE method to obtain the embedding of the adversarial candidate text $\boldsymbol{x}_{adv}$, which is denoted as $\boldsymbol{E}_{\boldsymbol{x}_{adv}}$. We transform the attacking task into a paraphrasing task. Specifically, $\boldsymbol{E}_{\boldsymbol{x}_{adv}}$ and $\boldsymbol{E}_{text}$ are concatenated as input data to predict whether the attack is successful (label 1) or not (label 0). The pseudo code of the algorithm is presented in Alg. 2.

**TPPEP Searching Algorithm**. After training the TPPEP model $f_p$, we consider all candidate adversarial texts $\boldsymbol{x}_{adv}$ of input text $\boldsymbol{x}$ and calculate the embedding $\boldsymbol{ED}$ of both $\boldsymbol{x}_{adv}$ and $\boldsymbol{x}$. We then apply the TPPEP method to $\boldsymbol{ED}$ and calculate the score of the successful attack. The adversarial candidate text with the highest paraphrasing score calculated by the TPPEP method is chosen to deploy the attack.

## A.4 Defense Method

We have initiated a comprehensive discussion on defensive strategies to counter punctuation-level attacks. In practical systems, we thoroughly investigate various defense approaches, including pre-completion and post-completion of training.

**Algorithm 2** TPPEP Training

---

**Input:** The training data $\boldsymbol{D} = \left\{ \left(\boldsymbol{x}^1, \boldsymbol{x}^1_{adv}, y^1_{att}\right), \left(\boldsymbol{x}^2, \boldsymbol{x}^2_{adv}, y^2_{att}\right), \cdots, \left(\boldsymbol{x}^N, \boldsymbol{x}^N_{adv}, y^N_{att}\right) \right\}$. The $\boldsymbol{x}^i$
  is input text, the $\boldsymbol{x}^i_{adv}$ is adversarial candidate text, and $y^i_{att}$ is the result of attacking (successful
  attacking is denoted as label 1; else denoted as label 0). The max train epoch $e_{max}$, the substitute
  model $f_{sub}$, the embedding model $TPPE$
**Output:** The trained TPPEP model $f_p$
  **for** $i = 1$ to $N$ **do**
    $\boldsymbol{E}^i_{text} = f_{sub}(\boldsymbol{x}^i)$
    $\boldsymbol{E}^i_{\boldsymbol{x}_{adv}} = TPPE(\boldsymbol{x}^i_{adv})$
    The input embedding $\boldsymbol{E}^i = concat(\boldsymbol{E}^i_{text}, \boldsymbol{E}^i_{\boldsymbol{x}_{adv}})$
  **end for**
  The embedding of training data $\boldsymbol{ED} = \left\{ \left(\boldsymbol{E}^1, y^1_{att}\right), \left(\boldsymbol{E}^2, y^1_{att}\right), \cdots, \left(\boldsymbol{E}^N, y^N_{att}\right) \right\}$
  **for** $i = 1$ to $e_{max}$ **do**
    // Train $f_p$ on $\boldsymbol{ED}$ to adjust the parameters $\theta_{f_p}$
    $\theta_{f_p} \leftarrow \text{train}(f_p, \boldsymbol{ED})$
  **end for**
  $f_p = f_p(\boldsymbol{ED}; \theta_{f_p})$
  **return** $f_p$

---

Table 10: The results of Preceding Language Modifier

| | | Without PLM | | | With PLM | | |
|---|---|---|---|---|---|---|---|
| | mode | TOP-1 | TOP-3 | TOP-5 | TOP-1 | TOP-3 | TOP-5 |
| ELECTRA | Insertion | 28.76% | 52.64% | 63.57% | 20.81% | 28.19% | 32.79% |
| | Displacement | 43.05% | 60.12% | 76.03% | 22.16% | 33.33% | 39.61% |
| XLMR | Insertion | 67.40% | 73.06% | 73.83% | 23.39% | 33.75% | 39.31% |
| | Displacement | 36.05% | 66.35% | 73.44% | 20.69% | 25.49% | 30.02% |

### A.4.1 Preceding Language Modifier

In response to this challenge, we re-train the model using adversarial training, which can be pro-
hibitively costly and impractical. To address this issue, we have developed a modifier to restore the
attacked text as closely as possible to its original form. This modifier is created using a Seq2Seq
model trained using pairs of original and attacked texts. We have found this approach to be a viable
solution. So, we employ the extensive language model CoEdIT-XXL (11 billion parameters) to obtain
the modifier. For our experiments, we have chosen the susceptible CoLA dataset. The experimental
results, depicted in Table 10, showcase the effectiveness of the proposed modifier.

### A.4.2 Adversarial Training

Prior to model training, we simultaneously evaluate the outcomes of both adversarial training
techniques. Adversarial training is a machine learning technique that trains a model in the presence
of intentionally generated adversarial examples. The experimental results are presented in Table
11. Adversarial training has a limited impact on the accuracy of the text model. However, after
carrying out adversarial training, the model demonstrates improved robustness and achieves favorable
performance against punctuation-level attacks.

### A.5 Comparative Supplementary Experiment to Benchmark Methods

We conducted a comparative analysis involving Single-shot and single punctuation attack ($S^3P$) and
alternative attack methods. We selected TextFooler, Bert-attack, and DeepWordBug as benchmark
methods. We employed Fool Rate (%), Perturbed Words (%), Semantic Similarity, and Number
of Queries as evaluation metrics. Since $S^3P$ perturbs a single punctuation, we restricted the other
algorithms to focus on a single word but allowed multiple attacks on that word. The experimental
results are presented in Table 12. Notably, the $S^3P$ algorithm achieved state-of-the-art results in
terms of Fool Rate, Perturbed Words, Semantic Similarity, and Average Number of Queries. This
underscores the effectiveness of our punctuation-level attack strategy. In contrast to other algorithms,

Table 11: The results of adversarial training. "Displacement" is the fool rate of Displacement

| datasets | Accuracy | | Displacement | |
|---|---|---|---|---|
| | clean_train | adv_train | clean_train | adv_train |
| cola | 79.13% | 79.63% | 75.69% | 63.88% |
| qqp | 88.27% | 88.81% | 32.24% | 20.36% |
| wanli | 67.21% | 66.41% | 61.09% | 50.11% |
| average | 78.20% | 78.28% | 56.34% | 44.78% |

Table 12: The result of TPPEP and benchmark methords

| | ELECTRA | | | | XLMR | | | | |
|---|---|---|---|---|---|---|---|---|---|
| Cola | Fool Rate | Semantic Sim | Number of queries | Perturbed Words | Fool Rate | Semantic Sim | Number of queries | Perturbed Words | |
| Insertion | **67.40%** | 0.9919 | **1** | **0.00%** | 28.76% | 0.9925 | 1 | 0.00% | Label |
| Displacement | 36.05% | 0.9936 | 1 | 0.00% | **43.05%** | 0.9933 | **1** | **0.00%** | Label |
| Deletion | 5.18% | **0.9965** | 1 | 0.00% | 4.89% | **0.9965** | 1 | 0.00% | Label |
| Replacement | 24.64% | 0.9927 | 1 | 0.00% | 6.62% | 0.9894 | 1 | 0.00% | Label |
| Bert-attack | 14.38% | 0.9602 | 14.1 | 10.88% | 21.00% | 0.9616 | 14.3 | 10.88% | Score |
| DeepWordBug | 35.86% | 0.9752 | 12.1 | 10.88% | 31.93% | 0.9704 | 12.0 | 10.88% | Score |
| TextFooler | 18.60% | 0.9810 | 10.2 | 10.88% | 19.85% | 0.9854 | 10.2 | 10.88% | Score |
| Hossein | 28.57% | 0.9752 | 6.3 | 10.88% | 27.04% | 0.9855 | 8.1 | 10.88% | Score |
| Nora | 23.30% | 0.9954 | 11.1 | 10.88% | 13.61% | 0.9855 | 11.1 | 10.88% | Score |
| QQP | DistilBERT1 | | | | DistilBERT2 | | | | |
| Insertion | **14.72%** | **0.9978** | **1** | **0** | 8.67% | 0.9986 | 1 | 0 | Label |
| Displacement | 8.52% | 0.9966 | 1 | 0 | 7.21% | 0.9966 | 1 | 0 | Label |
| Deletion | 3.94% | 0.9975 | 1 | 0 | 5.06% | **0.9977** | 1 | 0 | Label |
| Replacement | 7.59% | 0.9955 | 1 | 0 | **16.70%** | 0.9972 | **1** | **0** | Label |
| Bert-attack | 9.19% | 0.9574 | 30.9 | 3.76% | 8.11% | 0.9574 | 28.9 | 3.76% | Score |
| DeepWordBug | 10.36% | 0.9849 | 24.6 | 3.76% | 10.88% | 0.9837 | 23.7 | 3.76% | Score |
| TextFooler | 8.08% | 0.9960 | 24.0 | 3.76% | 7.96% | 0.9890 | 22.8 | 3.76% | Score |
| Hossein | 3.68% | 0.9876 | 8.1 | 3.76% | 4.12% | 0.9877 | 9.3 | 3.76% | Score |
| Nora | 9.21% | 0.9916 | 27.9 | 3.76% | 11.56% | 0.9936 | 28.1 | 3.76% | Score |
| Wanli | RoBERTa | | | | DeBERTa | | | | |
| Insertion | 8.44% | 0.9936 | 1 | 0 | 15.28% | 0.9910 | 1 | 0 | Label |
| Displacement | 5.12% | 0.9980 | 1 | 0 | 10.28% | 0.9978 | 1 | 0 | Label |
| Deletion | 3.22% | **0.9988** | **1** | **0** | 5.74% | **0.9987** | **1** | **0** | Label |
| Replacement | 8.48% | 0.9942 | 1 | 0 | 6.92% | 0.9967 | 1 | 0 | Label |
| Bert-attack | 23.28% | 0.9686 | 33.3 | 3.10% | 26.95% | 0.9456 | 35.6 | 3.10% | Score |
| DeepWordBug | 12.48% | 0.9724 | 23.3 | 3.10% | 14.43% | 0.9795 | 23.2 | 3.10% | Score |
| TextFooler | 20.58% | 0.9884 | 24.4 | 3.10% | 17.10% | 0.9958 | 24.5 | 3.10% | Score |
| Hossein | 7.20% | 0.9885 | 7.1 | 3.10% | 5.96% | 0.988853 | 8.2 | 3.10% | Score |
| Nora | 4.12% | 0.9876 | 33.3 | 3.10% | 4.12% | 0.987668 | 33.3 | 3.10% | Score |

our approach achieves higher fooling rates through zero-word perturbations, single-query attacks, improved semantic retention, and reduced perceptual impact.

### A.5.1 Analysis of Punctuation-Level Attacks

Our analysis has focused on the factors influencing the Fool Rate of punctuation-level attacks. In the main part, we emphasize the importance of vectorizing post-adversarial text by incorporating information about punctuations and their positions. Therefore, we investigate how the combination of position and punctuation information influences the average Fool Rate. Table 13 presents the impact of different punctuation mark types on the Fool Rate, listing the eight punctuation marks with the highest average deception rates. Remarkably, the "?" punctuation mark demonstrates the highest average Fool Rate, an impressive 13.27%. This suggests that, when taking into account the Cola, QQP, and Wanli datasets, along with their corresponding six models, for a specific sample subject to iterative insertion, deletion, replacement, or displacement of the "?" punctuation mark, the average Fool Rate is 13.27%.

In Table 8, "preceding" is used to refer to the initial one-third of the positions in the sentence, while "subsequent" indicates the final one-third of the position configurations in the sentence. With respect to insertion attacks, optimal results are achieved through the central segment of the inserted sentence. As for replacement and displacement attacks, heightened effectiveness is observed in the subsequent portion of the attacked sentence.

### A.6 Attack Convergence Discussion

The outcomes of multiple attacks are presented in Table 9. We observe a significant increase in the fooling rate of the model following multiple attacks. The fool rates are gradually converging. Additionally, we noted that beyond a certain threshold of attack iterations, the magnitude of the fooling rate enhancement becomes relatively stable. In such cases, we posit that it is appropriate to proceed with the next punctuation-level attack process.

Table 13: The results of the effect of punctuation on the fooling rate

| Punctuation | ? | , | ? | : | , | ! | . | ! |
|---|---|---|---|---|---|---|---|---|
| Fool Rate | 13.27% | 13.11% | 11.77% | 10.93% | 10.57% | 10.29% | 10.08% | 9.37% |

## A.7  Broader Impacts and Discussions

We are currently expanding the applicability of both TPPE and TPPEP methods to various NLP tasks. The promising performance across multiple tasks, including tasks like TC, paraphrasing, NLI, sss, summarization, and T2I, has instilled optimism regarding the future of punctuation-level attacks.