# OpenReview forum: "Punctuation-level Attack: Single-shot and Single Punctuation Can Fool Text Models"
_NeurIPS.cc/2023/Conference — NeurIPS 2023 poster_

### Official Review · Reviewer_rnAT · 2023-06-17

**Soundness:** 3 good
**Presentation:** 3 good
**Contribution:** 2 fair
**Rating:** 4
**Confidence:** 5

**Summary:**

The paper proposes a novel mode of textual attack, punctuation-level attack, which aims to fool text models by conducting punctuation-level perturbations, including insertion, displacement, deletion, and replacement. This paper also introduces Text Position Punctuation Embedding (TPPE) as an embedding method and Text Position Punctuation Embedding and Paraphrase (TPPEP) as a search method to reduce the search cost and time complexity of determining optimal positions and punctuation types for the attack. The paper demonstrates the effectiveness of the punctuation-level attack and the proposed methods on various tasks such as summarization, semantic-similarity scoring, and text-to-image tasks.

**Strengths:**

1. The proposed punctuation-level attack expands the scope of adversarial textual attacks. By focusing on punctuation-level perturbations, the authors provide an approach to fooling text models while minimizing its impact on human perception.
2. The proposed methods, TPPE and TPPEP, not only enhance efficiency but also reduce computational costs. Additionally, the authors present a comprehensive mathematical analysis of these approaches.
3. The effectiveness and versatility of the proposed punctuation-level attack are demonstrated by the experimental results on various datasets and state-of-the-art (SOTA) models.


**Weaknesses:**

1. Whether LLMs can also be fooled, which should be discussed in the paper

I guess such robustness is due to the amount of training data is not large enough. It is curious that LLMs like ChatGPT will still fall into such a deficiency, since LLMs are trained on huge data.

2. Why PLMs fail on punctuations is not discussed

The punctuation-level attack does not surprise me a lot. The most interesting problem is to probe into the reason why PLMs can be fooled by punctuations.
Unfortunately, this is not in the paper, which large limits the contribution of the work.

3. The defense is not discussed

The authors do not provide the study on how to defense the punctuation-level attack. The contribution on how to enhance language models e.g. DeBERTa to be robust against punctuation attacks is much more significant than how to attack. To me, this is very important for the community to improve real-world systems against underlying attackers.

4. The attack success rate is only promising on CoLA, while limited on the other datasets.
However, CoLA is not suitable dataset for evaluation.

CoLA requires the model to decide whether the given sentence is linguistically correct. The manipulation of punctuations can spoil the label of the original sentence.

5. Punctuation modifications can change the original label on some task

Punctuation-level attack is safer than word-level modification. It still can change the original label on some task, e.g. CoLA. Even in extreme cases, punctuations can change the entire semantics. It is better for the authors to discuss this part in the paper, e.g. some bad cases. This is also important for future research.


**Questions:**

How do you get the E_text, and E_punc? Do you use pretrained models or a finetuned one on the target task? The embeddings you extract are from the embedding layer or the last layer of the transformer blocks?

For E_punc, do you just input a single punctuation as the complete sentence into the model? If so, this is quite strange.

What are the dimensionalities for E_text, E_pos and E_punc and how can they be concatenated or added up?

You mentioned “We transform the attacking task into a paraphrasing task.” In line 269 when introducing TPPEP. What is the model used to conduct this paraphrasing task?

You mentioned “The adversarial candidate text with the highest paraphrasing score calculated by 272 the TPPEP method is chosen to deploy the attack.” In line 271-272. It seems that you need to do several queries and give them a rank. Is this contradictory to the O(1) complexity you claimed?


**Limitations:**

see the weakness part

---

> ### Author Rebuttal · Authors · 2023-08-10
>
> Q1: Whether LLMs can also be fooled
>
> A1: Due to time constraints, we focused our efforts on the summarization task using ChatGPT, limiting our use to a mere 20 samples per task. Our  attack method involved the insertion of individual punctuation. Following the application of this attack strategy, the ROUGE-1 score exhibited a decrease from 22.8 to 16.7. Despite this decline, we assert that punctuation-level attacks remain effective for complex tasks such as generating  task. Our experimental outcomes illustrate the vulnerability of even extensively trained Stable-diffusion models to punctuation-based attacks. Additionally, existing literature has already demonstrated the susceptibility of LLMs to adversarial attacks on textual data.
>
>
>
> Q2: Why PLMs fail on punctuations
>
> A2: Analying the post-attack data distribution highlights the text altered by punctuation-based attacks as an Out-of-distribution (OOD) dataset for pre-trained language models (PLMs). For instance, in the Replacement attack method, instances involve replacing '!' with '！', '?' with '？', and ',' with '，'. Given the rarity of Chinese punctuation marks in the training data of English-based PLMs, sentences transformed by punctuation-based attacks become OOD samples for PLMs. Consequently, this contributes to reduced generalization performance of PLMs on such data. This aspect underlies the ineffective punctuation handling by PLMs.
>
> In practical applications of PLMs, the prevailing approach involves training PLMs through fine-tuning. During the fine-tuning phase for downstream tasks, the training datasets for these tasks exhibit a lower proportion of punctuation errors. In theory, the finetuning process for downstream tasks could render the model more susceptible to OOD attacks, thereby further compromising the generalization capacity of PLMs for sentences that have undergone punctuation-level attacks.
>
>
>
> Q3: The defense is not discussed
>
> A3: In our global rebuttal, we have incorporated experimental outcomes to supplement our defense strategy.
>
> We selected Preceding Language Modifier and  adversarial training  as defense.  The efficacy of the Preceding Language Modifier and adversarial training are demonstrated in Table 6 and Table 8.
>
>
>
>
>
> Q4: The attack success rates  limited.
>
> The experimental results presented in this paper are obtained under highly challenging conditions, specifically the most extreme scenarios within the experiment setup. For instance, the experiments involve perturbing words to zero, conducting single punctuation attacks, allowing only a single query, and considering models that provide hard-label outputs in black-box attack scenarios. This deliberate choice of experimentation parameters has led to constrained outcomes. In Table 8, we have exclusively constrained the number of perturbed words for other benchmark methods. Despite the potential for other benchmark methods to access the model multiple times, obtaining posterior probabilities from black-box model outputs, and even in the context of white-box attacks, our TPPEP algorithm has demonstrated superior performance in terms of fool rate. Notably, across metrics including Perturbed Words (%), Semantic Similarity, and Number of Queries, TPPEP has consistently achieved wonderful  results across the entire dataset when compared to these benchmarks.
>
> We have also investigated the impact of increasing the number of model queries on the fool rate. Our findings reveal that when the  number of qurried increase to five queries (TOP-5), TPPEP achieves state-of-the-art misclassification rates across all datasets.
>
>
>
>
>
> Q5: Punctuation modifications can change the original label and the entire semantics on some task.
> In the revised edition, we would employ other text classification datasets to substitute cola
> In Table 8, we delve into the alterations in semantics resulting from punctuation-level attacks. We computed an average semantic similarity of 0.995526417 before and after punctuation attacks. The specific outcomes are in Table 8. Our current experimental findings suggest that our punctuation-level attack algorithm has a limited impact on semantics.
> In the revised version, we will incorporate modifications to prevent the emergence of extreme cases. Specifically, for the set of perturbed candidate texts, we will introduce a semantic similarity threshold. Any candidate text falling below this threshold will be excluded from consideration as a potential attack candidate.
>
>
>
>
>
> Q6: Questions about  E_text, and E_punc
>
> We trained a substitute model and employed the embeddings of its final layer as the output. For both individual text and standalone punctuation inputs, we fed them into the substitute model, extracting the outcomes from its final layer, denoted as E_text and E_punc, respectively.
>
>
>
> Given the pivotal role that punctuation types and placements play in attacks at the punctuation level, we sought to underscore the significance of punctuation. Consequently, we deliberately employed a singular punctuation mark as an entire sentence input into the model.
>
>
>
> The dimensions of E_text, E_pos, and E_punc are all set to 50. We opted for the addition method.
>
>
>
> Our TPPEP method draws inspiration from paraphrasing tasks, which involve assessing the relationship between two given texts. In the case of TPPEP, we leverage pre- and post-adversarial attack embedding vectors to determine the success of the attack.
>
>
>
> During the testing phase, a single access to the substitute model is sufficient to obtain the ranking of the perturbed text, which indicates our time complexity of O(1).
>
> Further elaboration on the embedding process and complexity is provided in the supplementary materials. Due to the word limitations, please  refer to our supplementary materials for additional details.

---

### Official Review · Reviewer_AjFS · 2023-07-03

**Soundness:** 3 good
**Presentation:** 3 good
**Contribution:** 4 excellent
**Rating:** 7
**Confidence:** 4

**Summary:**

By proposing a new type of adversarial attacks, i.e., the punctuation-level attacks, This paper can fool text models with less impact on the semantic information by human beings understanding. Its effectiveness is verified by experimental results on various datasets/tasks and victim models. What’s more, the attack method is accelerated by the proposed Text Position Punctuation Embedding and Paraphrase (TPPEP) approach, so that the attack can be accomplished with constant time cost. The efficiency has been demonstrated by the experimental studies.


**Strengths:**

+ This paper is well written and easy to follow.

+ This reviewer appreciates the novelty of the proposed punctuation-level attacks, and the motivation behind, i.e., attacking text models with minimal perceptual influence on human eyes. It indeed brings some insights.

+ Besides its effectiveness in fooling various textual models, the authors also propose a TPPEP method to accelerate the attacking procedure. I believe this can significantly improve the practical use of the punctuation-level attacks.

+ The attack results on the update-to-date Stable Diffusion model are quite interesting.


**Weaknesses:**

- Though the proposal of punctuation-level attacks is indeed well motivated, an obvious major concern is raised that only three tasks (thus three types of victim models) are selected for attack effectiveness evaluation in the experimental studies. It would be better if more methods are selected for evaluation. The proposed approach would be fully validated and the conclusion would be more convincing.

- The current experimental results fail to provide more insights on how the method work in different scenarios. In fact, as a new type of text-attack method, there should be more in-depth analyses and explanations to quantitatively or qualitatively show the effectiveness.

- The notations in the paper are not always consistent. And some parts are not well illustrated, especially in Sec. 3.3. It is suggested to be re-organized.


**Questions:**

As I stated, the results of text-to-image task are interesting, while it is not so convincing. At the least, how can we know if the attack is successful or not, by human eyes? There should be more convincing validation.

Overall, I like the idea in the paper and the current drawback is the results. Please consider my suggestions about the experimental studies. I am open to increasing my score if my concerns are satisfactorily addressed.


**Limitations:**

The current discussion about the limitations is practical but not so comprehensive. For example, the resource required by the method is not discussed.

---

> ### Author Rebuttal · Authors · 2023-08-09
>
> Q1: It would be better if more methods are selected for evaluation.
>
> A1: Tables B, C, and D present the results of our endeavor to extend punctuation-level attacks, TPPE, and TPPEP algorithms to encompass three distinct tasks: summarization, semantic similarity scoring, and text-to-image generation.
>
> In the context of the summarization task, we employ the ROUGE-1 metric as the dependent variable y and formulate a predictive model to estimate the ROUGE-1 score for the test set, and subsequently, we select the candidate attack text characterized by the lowest predicted ROUGE-1 value.
>
> For the text-to-image task, we utilize the CLIP score as y and identify the candidate attack text by determining the text associated with the smallest predicted CLIP score value.
>
> In the semantic similarity scoring task, our y corresponds to the semantic similarity between the text before and after the attack. To model this, we develop a predictive model to estimate the semantic similarity values for the test set.
>
> ​                                                   Table B  The results of text to image task
>
> |            | Without Attack | TOP-1  | TOP-3  | TOP-5  | Traveral |
> | ---------- | -------------- | ------ | ------ | ------ | -------- |
> | clip-score | 0.3278         | 0.3176 | 0.3069 | 0.3022 | 0.2610   |
>
> ​                                                      Table C  The results of summarization
>
> |         | Without Attack | TOP-1 | TOP-3 | TOP-5 | Traveral |
> | ------- | -------------- | ----- | ----- | ----- | -------- |
> | ROUGE-1 | 11.69          | 10.91 | 9.65  | 9.11  | 5.22     |
>
> ​                                          Table D  The results of semantic-similarity-scoring
>
> |                | Sentence-BERT | Distilbert |         |          |
> | -------------- | ------------- | ---------- | ------- | -------- |
> | ST12           | Peaeson       | Spearman   | Peaeson | Spearman |
> | Without Attack | 0.7990        | 0.6988     | 0.8056  | 0.7257   |
> | TOP-1          | 0.7874        | 0.6862     | 0.7902  | 0.7035   |
> | TOP-3          | 0.7760        | 0.6738     | 0.7759  | 0.6990   |
> | TOP-5          | 0.7654        | 0.6626     | 0.7649  | 0.6745   |
> | Traveral       | 0.6992        | 0.5832     | 0.6694  | 0.6048   |
>
> The attack methodologies showcased in Tables B, C, and D predominantly encompass insertion-based techniques. Due to temporal constraints, we intend to incorporate the outcomes of alternative attack approaches in the revised version of this paper.
>
> Moreover, in order to comprehensively assess the effectiveness of our attack algorithm, we have included several additional evaluation metrics in Table 8 of the global rebuttal. In addition to the fooling rate, we have incorporated metrics such as Perturbed Words, Semantic Similarity, and Average Number of Queries. At present, our TPPEP algorithm has established state-of-the-art results across these metrics.
>
>
>
> Q2:  As a new type of text-attack method, there should be more in-depth analyses and explanations to quantitatively or qualitatively show the effectiveness.
>
> A2: In the  global rebuttal, we have included discussions on defense mechanisms, comparative outcomes with other algorithms, and more. Our intent is for these experimental findings to offer additional insights into punctuation-level attacks.
>
>
>
> Q3: The notations in the paper are not always consistent.
>
> A3:  We greatly appreciate your valuable suggestions. We will meticulously review the notations throughout the entire manuscript. In the revised version, we will also reorganize Section 3.3 in accordance with your recommendations.
>
>
>
>
>
> Q4:As I stated, the results of text-to-image task are interesting, while it is not so convincing.
>
> In the global rebuttal, we present the latest experimental outcomes for the text-to-image task. We utilize the CLIP score as the metric for evaluating attacks and leverage the Pokemon Blip Captions dataset, splitting it into training and testing sets in an 8:2 ratio.
>
> In the supplementary materials of the global rebuttal, we present the latest experimental findings pertaining to the text-to-image domain. Employing the clip-score as the metric for assessing the attacks, the outcomes within Table 3 elucidate the impact of attacks conducted at the punctuation level. Notably, following these attacks, the clip-scores experienced a reduction from 0.3281 and 0.4040 to 0.2484 and 0.3468, respectively.

---

> > ### Comment · Reviewer_AjFS · 2023-08-17
> > **Thanks for the rebuttal**
> >
> > I appreciate the effort in providing the rebuttal. In general, I think my concerns are well solved, especially the experimental study issue. The paper quality will be improved by addressing the concerned issues in the revision. I have raised my rating a bit accordingly.

---

> > > ### Author Response · Authors · 2023-08-17
> > > **Thanks for the response**
> > >
> > > Thanks a lot for your pertinent feedback and suggestions, and your reconsideration of our work. We will take all the suggestions/requests from all the reviewers/chairs into consideration to make the paper more solid.

---

### Official Review · Reviewer_y7qU · 2023-07-05

**Soundness:** 3 good
**Presentation:** 3 good
**Contribution:** 3 good
**Rating:** 7
**Confidence:** 4

**Summary:**

This paper introduces a new approach to textual attacks called the punctuation-level attack. The method aims to fool text models while minimizing its impact on human perception and understanding. The paper discusses the effectiveness of this attack strategy, presents a search method to optimize its deployment, and provides experimental results showcasing its success. The authors also apply the single punctuation attack to summarization, semantic-similarity-scoring, and text-to-image tasks, achieving encouraging results. The paper concludes that the punctuation-level attack is more imperceptible to human beings and has less semantic impact compared to traditional character-/word-/sentence-level perturbations. The integrated Text Position Punctuation Embedding (TPPE) allows the punctuation attack to be applied at a constant cost of time. The experimental results on public datasets and state-of-the-art models demonstrate the effectiveness of the punctuation attack and the proposed TPPE.

**Strengths:**

1. The paper introduces a new approach to textual attacks called the punctuation-level attack, which is different from traditional character-/word-/sentence-level perturbations.
2. The punctuation-level attack is designed to be more imperceptible to human beings and has less semantic impact compared to traditional perturbations.
3. The paper presents a search method called Text Position Punctuation Embedding (TPPE) to optimize the deployment of the punctuation-level attack.
4. The paper provides experimental results showcasing the effectiveness of the punctuation-level attack and the proposed TPPE on public datasets and state-of-the-art models.



**Weaknesses:**

The adversarial attacks discussed in this paper can be categorized as non-pure white-box attacks, as the attack objective may differ from the model's evaluation metric. It is crucial to explicitly acknowledge this fact in the paper, as it is widely recognized that achieving white-box robustness represents an upper bound and is significantly more challenging than black-box robustness.

It appears that the number of attack iterations is restricted. To ensure robustness evaluation, it is advisable to ensure attack convergence by employing an adequate number of iterations.

In the experimental section, when comparing the proposed method with other approaches, a fixed number of updating steps is consistently utilized.



**Questions:**

The paper lacks specific information regarding the implementation details of adversarial training. It does not provide explicit explanations regarding the ratio of clean versus adversarial samples used during training, nor does it clarify whether all methods employ identical training strategies.

**Limitations:**

The main emphasis of the paper is placed on evaluating the punctuation-level attack's efficacy within specific tasks, including summarization, semantic-similarity-scoring, and text-to-image tasks. However, the evaluation of this attack is not comprehensive across a diverse set of NLP tasks, limiting the extent to which the findings can be generalized.

---

> ### Author Rebuttal · Authors · 2023-08-09
>
> Q1:The adversarial attacks discussed in this paper can be categorized as non-pure white-box attacks.
>
> A1:We greatly appreciate your suggestions, and we will incorporate the necessary revisions in the revised version of the paper. This aspect will be emphasized in the revised version as well.
>
>
>
> Q2:It appears that the number of attack iterations is restricted. To ensure robustness evaluation, it is advisable to ensure attack convergence by employing an adequate number of iterations.
>
> A2:The outcomes of multiple attacks are presented in Table 12. We observed a significant increase in the fooling rate of the model following multiple attacks. The fool rates are gradual convergencing. Additionally, we noted that beyond a certain threshold of attack iterations, the magnitude of the fooling rate enhancement becomes relatively stable. In such cases, we posit that it is appropriate to proceed with the next punctuation-based attack process.
>
> | Cola         |              | Top-10 | Top-20 | Top-30 |
> | ------------ | ------------ | ------ | ------ | ------ |
> |              | Insertion    | 74.59% | 75.17% | 76.51% |
> | ELECTRA      | Displacement | 77.89% | 78.93% | 79.41% |
> |              | Replacement  | 50.60% | 59.31% | 63.92% |
> |     XLMR        | Insertion    | 76.51% | 83.41% | 85.81% |
> |
> |              | Replacement  | 15.20% | 18.14% | 19.61% |
> | QQP          | Insertion    | 26.86% | 30.61% | 32.68% |
> | DistillBERT2 | Displacement | 23.46% | 26.16% | 26.67% |
> |              | Replacement  | 14.77% | 16.87% | 17.93% |
> | QQP          | Insertion    | 29.21% | 31.81% | 35.55% |
> | DistillBERT2 | Displacement | 19.88% | 22.50% | 23.09% |
> |              | Replacement  | 24.46% | 26.48% | 27.54% |
> |              | Insertion    | 37.02% | 45.36% | 49.60% |
> | RoBERTa      | Displacement | 15.96% | 20.41% | 23.07% |
> |              | Replacement  | 25.58% | 33.11% | 36.89% |
> |              | Insertion    | 44.70% | 53.00% | 56.00% |
> | deBERTa      | Displacement | 26.74% | 32.65% | 35.41% |
> |              | Replacement  | 22.51% | 29.34% | 33.97% |
>
> Q2:In the experimental section, when comparing the proposed method with other approaches, a fixed number of updating steps is consistently utilized.
>
> A3:Throughout the main text, supplementary materials, and in all experiments included in the global rebuttal submission, we consistently employed an identical number of updating steps.
>
>
>
>
>
> Q3: The paper lacks specific information regarding the implementation details of adversarial training.
>
> A4:In our adversarial training, we maintain a 1:1 ratio between clean and adversarial samples during the training process. Specifically, we employ a learning rate of 0.0002 for our adversarial training, spanning 27 epochs. The optimization process utilizes the cross-entropy loss function employing the Adam optimizer. It is noteworthy that all methodologies adhere to identical adversarial training strategies.
>
>
>
> Q4: The evaluation of this attack is not comprehensive across a diverse set of NLP tasks
>
> Tables B, C, and D present the results of our endeavor to extend punctuation-level attacks, TPPE, and TPPEP algorithms to encompass three distinct tasks: summarization, semantic similarity scoring, and text-to-image generation.
>
> In the context of the summarization task, we employ the ROUGE-1 metric as the dependent variable y and formulate a predictive model to estimate the ROUGE-1 score for the test set, and subsequently, we select the candidate attack text characterized by the lowest predicted ROUGE-1 value.
>
> For the text-to-image task, we utilize the CLIP score as y and identify the candidate attack text by determining the text associated with the smallest predicted CLIP score value.
>
> In the semantic similarity scoring task, our y corresponds to the semantic similarity between the text before and after the attack. To model this, we develop a predictive model to estimate the semantic similarity values for the test set.
>
> ​                                                   Table B  The results of text to image task
>
> |            | Without Attack | TOP-1  | TOP-3  | TOP-5  | Traveral |
> | ---------- | -------------- | ------ | ------ | ------ | -------- |
> | clip-score | 0.3278         | 0.3176 | 0.3069 | 0.3022 | 0.2610   |
>
> ​                                                      Table C  The results of summarization
>
> |         | Without Attack | TOP-1 | TOP-3 | TOP-5 | Traveral |
> | ------- | -------------- | ----- | ----- | ----- | -------- |
> | ROUGE-1 | 11.69          | 10.91 | 9.65  | 9.11  | 5.22     |
>
> ​                                          Table D  The results of semantic-similarity-scoring
>
> |                | Sentence-BERT | Distilbert |         |          |
> | -------------- | ------------- | ---------- | ------- | -------- |
> | ST12           | Peaeson       | Spearman   | Peaeson | Spearman |
> | Without Attack | 0.7990        | 0.6988     | 0.8056  | 0.7257   |
> | TOP-1          | 0.7874        | 0.6862     | 0.7902  | 0.7035   |
> | TOP-3          | 0.7760        | 0.6738     | 0.7759  | 0.6990   |
> | TOP-5          | 0.7654        | 0.6626     | 0.7649  | 0.6745   |
> | Traveral       | 0.6992        | 0.5832     | 0.6694  | 0.6048   |
>
> The attack methodologies showcased in Tables B, C, and D predominantly encompass insertion-based techniques. Due to temporal constraints, we intend to incorporate the outcomes of alternative attack approaches in the revised version of this paper.
>
> Moreover, in order to comprehensively assess the effectiveness of our attack algorithm, we have included several additional evaluation metrics in Table 8 of the global rebuttal. In addition to the fooling rate, we have incorporated metrics such as Perturbed Words, Semantic Similarity, and Average Number of Queries. At present, our TPPEP algorithm has established state-of-the-art results across these metrics.

---

### Official Review · Reviewer_LLfX · 2023-07-05

**Soundness:** 2 fair
**Presentation:** 2 fair
**Contribution:** 3 good
**Rating:** 5
**Confidence:** 3

**Summary:**

This paper introduces an adversarial attack against NLP models based on punctuation perturbations. The authors introduce an attack called Text Position Punctuation Embedding (TPPE) that comprises an insertion, displacement, deletion, and replacement attack based on textual punctuation (e.g., commas or periods).
Experiments are conducted on various datasets, ranging from text classification (CoLA) to paraphrasing (QQP) and natural language inference (WANLI). The attacks are applied to ELECTRA, XLMR, and BERT-based models (DistilBERT, RoBERTa, DeBERTa). Additionally, the attack is applied to semantic-similarity-scoring (STS12), summarization (gigaword), and text-to-image tasks (prompting Stable Diffusion V2). Experimental results are promising, showing that the attack can be used to successfully attack models for the above tasks.

**Strengths:**

* The paper provides an extensive analysis of punctuation-level attacks against NLP models. These attacks are promising since they have the potential to be less perceptible as compared to existing character-, word-, and sentence-level attacks.
* The analysis is extensive in that multiple NLP tasks (classification, summarization, text-to-image, etc.) are analyzed.
* It is interesting to see that the investigated models are vulnerable to punctuation-level attacks across tasks and domains.


**Weaknesses:**

* The paper does not compare TPPE against existing works and baselines. In Section 2, the authors point out various existing works focusing on punctuation attacks. However, none of these works have been evaluated and compared against in their experimental settings. To identify and support the strengths and utility of TPPE, such experiments are essential.
* I additionally think that comparisons to character-, word, and sentence-level attacks on the selected datasets would have been insightful since these experiments would provide the reader with a better understanding of how punctuation-level attacks perform in comparison to attacks focusing on other parts of a textual sequence.
* The paper does not further analyze the semantics of perturbed adversarial examples. To support the claims of semantic imperceptibility, human experiments analyzing the change in semantics between an original sequence and its adversarial counterpart would be important. The examples in Figure 2 nicely illustrate that inserting single punctuation marks can substantially impact the meaning of a sequence. Since adversarial examples are desired to preserve the semantics of an attacked sequence, quantitative experiments would be needed to evaluate TPPE in that context.
* The paper does not discuss potential approaches to mitigate the models’ vulnerability against punctuation-level attacks, for instance by assessing whether adversarial training / data augmentation (i.e., training the model on adversarially perturbed sequences) can help increase the robustness of the attacked models. This would provide additional insights into how robust the attack is, and how it can potentially be defended against.
* The results for the text-to-image task consist only of two qualitative examples. These examples are highly interesting, but to better evaluate the vulnerability of Stable Diffusion models against such attacks, quantitative results over a larger dataset would be important.
* Overall, the paper focuses too much on introducing the attack and discussing its details and time-complexity analysis, instead of extensively evaluating its performance (the Experiments Section spans under 2 pages in the manuscript).


**Questions:**

Could you provide a few additional details on model training / fine-tuning, as well as the dataset splits that you used for attacking the models? This is not mentioned in Section 4.1.

**Limitations:**

The paper briefly discusses Limitations in Section 5. However, potential ethical considerations arising from this research remain unaddressed. Since discussing these is quite important in this context (the proposed attack can be misused for malicious purposes), I would encourage the authors to add a section for this.

---

> ### Author Rebuttal · Authors · 2023-08-09
>
> Q1:Compare punctuation-level attack to the existing works focusing on punctuation attacks
>
> A1:First and foremost, it is imperative to clarify that the existing research concerning punctuation attacks does not pertain to the punctuation-level attack. The works referenced as [18, 16] are essentially char-level attacks. Their approach involves the insertion of punctuation marks within words and at word endings, leading to the transformation of the original word into an OOV word(e.g., "lo,ve" and "love,"). Similarly, the study denoted as [24] simply appends fifty tildes at the sentence conclusion,which has perturbed too much to be suitable as a benchmark method.
>
>  The comparative summary of our findings against the outcomes delineated in references [18, 16] is presented on Table 8 in the pdf of global rebuttal. “Hossein” and “Nota” are the method of [18,16] respectively. The results of Table 8 indicate TPPEP method is better than “Hossein” and “Nota” are the method.
>
>
>
> Q2:Compare punctuation-level attack to the char-, word-, sentence-level attacks
>
> The comparative results between our approach and the char- and word-level attack methods are presented in Table 8. The sentence-level attacks will attack the whole sentence, which has perturbed too much to be suitable as a benchmark method. The results of Table 8 indicate TPPEP method is better than other benchmark methods of char-level and word- level method.
>
>
> In this experiment, TPPEP perturbed only one punctuation mark. In order to make a fair comparison between TPPEP and other attack algorithms, we limit the other attack algorithms to only perturbing one word. So we didn't select SSTA[24] and sentence-level attacks as benchmarks.We have indeed computed the results of both SSTA [24] and the sentence-level method. If necessary, we can present these results in the discussion. Due to the word limitations,  we  do  not  present   here.
>
> Q3: Analyze the semantics of perturbed adversarial examples
>
> In Table 8 of  global rebuttal, we delve into the alterations in semantics resulting from punctuation-level attacks. We computed an average semantic similarity of 0.995526417 before and after punctuation attacks. The specific outcomes for each dataset, model, and attack technique are detailed under the "Semantic Sim" criterion in Table 8. Our current experimental findings suggest that our punctuation-level attack algorithm has a limited impact on semantics.
>
> In addition, we have incorporated human evaluations to assess the semantic similarity before and after attacks. We randomly selected 100 sentences along with 15 participants from each of the three datasets, alongside their respective perturbed texts. A five-point scale was established, where a score of 1 indicates substantial dissimilarity between the two sentences, and 5 indicates substantial similarity. The computed mean similarity score is 4.93.
>
>
>
>  Q4: The defense is not discussed
>
> In our global rebuttal, we have incorporated experimental outcomes to supplement our defense strategy. We introduce a defense technique based on the concept of a Preceding Language Modifier. The core objective of the Preceding Language Modifier is to restore the attacked text to its original, coherent form. Such rehabilitated text poses no threat to well-established models. The efficacy of the Preceding Language Modifier is demonstrated in Table 6, revealing its capacity to effectively mitigate the impact of punctuation-based attacks.
>
>
>
> We have also investigated the impact of adversarial training on punctuation attacks, and the experimental findings have been presented in Table 7. The results demonstrate that adversarial training effectively mitigates the effects of punctuation-level attacks without compromising the model's accuracy
>
>
>
> Q5: The results for the text-to-image task have not been discussed detailedly
>
> In  the global rebuttal, we present the latest experimental outcomes for the text-to-image task. We utilize the CLIP score as the metric for evaluating attacks and leverage the Pokemon Blip Captions dataset, splitting it into training and testing sets in an 8:2 ratio.
>
> We present the latest experimental findings pertaining to the text-to-image domain. Employing the clip-score as the metric for assessing the attacks, the outcomes within Table 9 elucidate the impact of attacks conducted at the punctuation level. Notably, following these attacks, the clip-scores experienced a reduction from 0.3281 and 0.4040 to 0.2484 and 0.3468, respectively.
>
>
>
> Q6: The paper focuses too much on introducing punctuation-level attack
>
> Diverging from the extensively explored domains of char-, word-, and sentence-level attacks, we introduce a pioneering approach focusing on attacks at the punctuation-level. So we believe it is necessary to details of the attack.
>
> We have conducted supplementary experiments, which are presented both in the global rebuttal section and earlier in this document.
>
> We greatly appreciate the insightful suggestions you provided regarding our experiments. We will effectuate revisions to the paper by incorporating additional experimental segments while judiciously trimming the content related to attack intricacies and time complexity discussions.
>
>
>
> Q7:Details of the experiment were not specified
>
> We have provided detailed information about training the substitute model in the supplementary materials (Section 1.2 in the supplementary materials). When training the TPPEP model, the training data was split into an 8:2 ratio for the training set and validation set, respectively. For training the TPPEP algorithm, a batch size of 300 was utilized over 9 epochs, employing the Adam optimizer with a learning rate of 0.001, and the loss function employed was the Cross-Entropy Loss.
>
> Q8：Potential ethical considerations arising from this research remain unaddressed
>
> We greatly appreciate your suggestions. In the revised version, we will incorporate a dedicated section to discuss potential ethical considerations.

---

> > ### Comment · Reviewer_LLfX · 2023-08-15
> > **Thanks to the authors!**
> >
> > I greatly appreciate the detailed response and provision of additional results! I have raised my score accordingly.
> >
> > However, given that the authors added a large number of additional results which would need to be incorporated into the manuscript (and will hence likely change central parts of the paper), I believe that this paper could also benefit from an additional round of reviews. I am happy to further discuss this with other reviewers and ACs.

---

> > > ### Author Response · Authors · 2023-08-16
> > > **The central part does not change**
> > >
> > > Dear Reviewer LLfX,
> > >
> > > Thanks a lot for your reply and your acknowledgment of our effort in the rebuttal. And we also appreciate your updated evaluation of our work.
> > >
> > > In terms of your concern, we want to emphasize that, the most significant contribution of this paper, is still the first proposal of the punctuation-level attack and its associated search method to accelerate the attack. So, the central part does not change even considering the new results and discussion in the rebuttal.
> > >
> > > We appreciate the insightful feedback from all reviewers, and we will accordingly update the manuscript. Primarily, we will consider the necessary discussion in terms of defense and in-depth analysis of punctuation-level attacks, and the most important results in the revision. Due to page limitations, all the other materials and experimental results will be included in the Appendix. We hope this solves your concern well.

---

### Author Rebuttal · Authors · 2023-08-09

Upon receiving the reviewers' feedback, several sections have been incorporated to refine the original paper.

# **1 Defense Methord**



We have introduced an in-depth discourse concerning defense strategies aimed at countering

punctuation-level attacks. In the realm of real-world systems, we meticulously investigate a range

of defense approaches both pre- and post-completion of training.

## 1.1 **Preceding Language Modifier**

For after-trained models, initiating a retraining process with adversarial training methods is evidently costly and impractical. To address this scenario, we propose a modifier that aims to restore the text to its original form as much as possible following punctuation-level attacks. An alternative approach that involves training a seq2seq model using both punctuated and original texts post-attack could be considered. However, due to time constraints, we employ prompt learning using the grammatically enhancing large language model CoEdIT-xxl (11 B) as the modifier. We opt to conduct experiments on the most vulnerable CoLA dataset and the two strongest attack mode , e.g. Insertion and  Displacement. The experimental outcomes, as depicted in Table 6, demonstrate the favorable performance of our modifier strategy. As time limitations apply, results for other datasets will be presented in the revised version of this work.

## **1.2 Adversarial Training**

For untrained models, initiating a retraining process with adversarial training methods is evidently costly and impractical. To address this scenario, we propose a modifier that aims to restore the text to its original form as much as possible following punctuation-level attacks. An alternative approach that involves training a seq2seq model using both punctuated and original texts post-attack could be considered. However, due to time constraints, we employ prompt learning using the grammatically enhancing large language model CoEdIT-xxl (11 B) as the modifier. We opt to conduct experiments on the most vulnerable CoLA dataset. The experimental outcomes, as depicted in Table 6, demonstrate the favorable performance of our modifier strategy. As time limitations apply, results for other datasets will be presented in the revised version of this work.

# 2  Comparision to Benchmarks
We conducted a comparative analysis against alternative attack algorithms, namely TextFooler, Bert-attack, and DeepWordBug, which were chosen as benchmarks.And Fool Rate  , Perturbed Words (%) , Semantic Sim, and Number of queries are Selected as evaluation metrics.  T5-large pre-train model is selected as the tool to caculate Semantic Sim. And the “Label” indicate the output of query is hard-label, and “score” indicate the output of query is predict probabilities.

 As the TPPEP algorithm perturbs solely a single punctuation mark, for the purpose of a fairer comparison, we imposed the constraint that other algorithms are also allowed to target only one word, albeit with the ability to attack this word multiple times. The experimental findings, presented in Table 8, demonstrate that the TPPEP has consistently achieved state-of-the-art results across various metrics, including Fool Rate, Perturbed Words, Semantic Similarity, and Average Number of Queries. Evidently, our approach at the punctuation level attains improved fool rates in scenarios involving zero-word perturbation, single-query access, preservation of semantic information, and reduced perceptibility, when compared to alternative algorithms.

# 3 **Experiments On Text-to-Image Task**

We have expanded the quantitative experimentation section for T2I applications, employing the clip-score as our evaluation metric. The experimental outcomes are presented in Table 9, where "ori-text1" refers to the sentence "a corgi is playing piano, oli on canvas". Upon inserting a comma for punctuation-level attack, "adv-text1" is obtained from "ori-text1". And "ori-text0" refers to the sentence “a professional photograph of an astronaut riding a triceratops”Notably, with a high semantic similarity of 0.9843 between "ori-text1" and "adv-text1", the clip-score decreases from 0.4040 to 0.3468 post-attack.

Additionally, in order to assess the robustness of the Stable Diffusion model against punctuation-level attacks, we employ the "pokemon-blip-captions" dataset for evaluating the attack's impact. This dataset comprises a total of 832 texts in its training and testing subsets. Following punctuation insertion attacks, the overall clip-score declines from 0.3273 to 0.2590. This reduction is further reflected in both the training and testing subsets, underscoring the model's vulnerability.

# 4 Analysis of Punctuation-Level Attacks

We examine the factors influencing the success rate of punctuation-based attacks. In our textual adversarial perturbations, punctuation information and its positional context are essential components. Thus, we delve into the impact of positional information and punctuation characteristics on the average Fool Rate.

Table 10 illustrates the effects of various punctuation types on the Fool Rate, detailing the top eight punctuation marks with the highest average misclassification rates. The "?" symbol emerges as the most influential, exhibiting an elevated average misclassification rate of 8.36%. This signifies that, across the cola, QQP, and wanli datasets, as well as their respective six models, when the "?" symbol is iteratively inserted, deleted, replaced, or displaced in a specific sample, the average probability of successful fooling reaches 8.36%.

In Table 11, the term "preceding" pertains to the initial one-third position of the sentence, while "subsequent" refers to the final one-third position of the sentence. Concerning insertion attacks, the most effective outcome is observed when the middle portion of the inserted sentence is targeted. In the context of replace and displace attacks, the subsequent segment of the attacked sentence yields optimal results.

---

### Decision · Program_Chairs · 2023-09-21

**Decision:**

Accept (poster)

**Comment:**

I will suggest acceptance for this work, mainly because reviewers appreciated the paper’s extensive analysis (LLfX) and the novelty of the punctuation based approach (y7qU, AjFS). However, my certainty on this paper is somewhat lower than usual.

Questions were raised in the reviewing period regarding baselines and comparisons to other kinds of attacks (LLfX), perturbation semantics (LLfX), strength of evaluation of the methods (LLfX, y7qU, AjFS), number of attack iteration (y7qU). Some reviewers also mentioned that ideas for mitigations would have been interesting to include or discuss (LLfX, rnAT), as would LMs (rnAT)---this AC agrees that these would be interesting, but their non-inclusion does not warrant rejection in my view (although it would probably be prudent to remove references to ChatGPT since that model isn’t addressed).

In the rebuttal, authors shared new comparisons to other attacks and perturbation semantics (with human eval). Authors also discussed the number of attack iterations in the rebuttal, and these results suggested convergence. Authors also offered some defense against punctuation attacks. The amount of additional work presented in rebuttal led LLfX to suggest another round of review might be warranted, but they did not change their score.

One reviewer (rnAT) was an outlier on the low end, and they were very confident. Although I agree with them that the choice of tasks was a bit unusual and that label change, particularly on CoLA should be mentioned, several of their suggestions outside the scope of the work in my opinion (e.g. “The punctuation-level attack does not surprise me a lot. The most interesting problem is to probe into the reason why PLMs can be fooled by punctuations. Unfortunately, this is not in the paper, which largely limits the contribution of the work.”).